# Enhancing Cancerous Gene Selection and Classification for High-Dimensional Microarray Data Using a Novel Hybrid Filter and Differential Evolutionary Feature Selection

**DOI:** 10.3390/cancers16233913

**Published:** 2024-11-22

**Authors:** Arshad Hashmi, Waleed Ali, Anas Abulfaraj, Faisal Binzagr, Entisar Alkayal

**Affiliations:** 1Department of Information Systems, Faculty of Computing and Information Technology, King Abdulaziz University, P.O. Box 344, Rabigh 21911, Saudi Arabia; 2Information Technology Department, Faculty of Computing and Information Technology, King Abdulaziz University, P.O. Box 344, Rabigh 21911, Saudi Arabia; ealkayyal@kau.edu.sa; 3Department of Computer Science, Faculty of Computing and Information Technology, King Abdulaziz University, P.O. Box 344, Rabigh 21911, Saudi Arabia; fbinzagr@kau.edu.sa

**Keywords:** cancer classification, differential evolutionary algorithm, filter feature selection, gene selection, high-dimensional microarray dataset, cancer classification, brain cancer, breast cancer, lung cancer, central nervous system cancer

## Abstract

To improve cancer classification performance for high-dimensional microarray datasets, this work proposes combining filter and differential evolutionary (DE) algorithm feature selection techniques. By scoring genes or features of high-dimensional microarray datasets by some common filter methods, we keep only the highest-ranked features and eliminate superfluous and irrelevant ones to decrease the dimensionality of the microarray datasets. Then, the genes or features of the microarray datasets are optimized further by DE, producing noticeably better classification results. This could lead to outstanding improvement in the cancer classification using only less features of the microarray datasets.

## 1. Introduction

Cancer research is widely acknowledged as a highly promising domain for using machine learning. Extensive endeavours have been undertaken to explore prospective approaches for detecting and treating cancer [1].

Cancer is a condition that exhibits uncontrolled cellular proliferation and results as a growth of a tumor in the form of mass or lump. Lung, colon, breast, central nervous system (CNS), liver, kidney, prostate, and brain cancer are among the various types of cancer that can occur. In this research study we have examined four distinct types of cancer dataset: Lung, Breast, Brain, and Central Nervous System. Lung cancer is a prevalent and mortal cancer worldwide [2]. It can arise in the primary airway, specifically within the lung tissue. The outcome is the unregulated proliferation and growth of specific lung cells. Respiratory disorders, including emphysema, are linked to an increased risk of lung cancer development. Breast cancer is one of the most invasive malignancies, predominantly affecting women. It is considered the most severe cancer following lung cancer due to the elevated mortality rate among women [3,4]. The rapid development of abnormal brain cells that is indicative of a brain tumor [5,6,7] is a significant health concern for adults, as it can result in severe impairment of organ function and even mortality. A malignant brain tumour rapidly grows and extends to adjacent brain regions. The Central Nervous System (CNS), consisting of the brain and spinal cord, is responsible for numerous biological functions. Spinal cord compression and spinal instability often involve the vertebral and spinal epidural spaces as common sites for cancer metastases. Metastases represent the most common type of CNS tumour in adults [8].

Cancer is regarded as one of the primary causes of death. In order to preserve the lives of patients, advanced technologies such as artificial intelligence and machine learning are used to detect cancer at an early stage and accurately predict its type. The cancer diagnosis is performed by employing several medical datasets, which encompass microarray gene expression data, also known as the microarray dataset. Microarray technology offers unique experimental capabilities that have been beneficial to cancer research. Microarray data can be used to evaluate a wide variety of cancer types. High-dimensional data from DNA microarray experiments is known as gene expression data. It is widely used to classify and detect malignant disorders [9]. The most recent development of artificial intelligence, specifically machine learning, has simplified data analysis, including microarray data. The authors [10] demonstrated that machine learning algorithms can be employed for microarray dataset analysis for cancer classification. Utilizing expressions of genes in microarray datasets can serve as an effective tool for diagnosing cancer. However, the number of active genes continues to grow, surpassing hundreds of thousands, while the available datasets remain limited in size, containing only a few subsets of samples. Therefore, one of the challenges in analyzing microarray datasets used for cancer classification is the curse of dimensionality. There is an additional concern regarding the characteristics of the current microarray datasets, which consist of numerous redundant and irrelevant features that have a detrimental impact on cancer classification results and computational expense [11]. The presence of duplicated and irrelevant features in very high-dimensional microarray datasets reduces the ability of the machine learning techniques to achieve accurate cancer classification and prediction [12]. These characteristics diminish the efficiency of the prediction model and complicate the search for meaningful insights. Consequently, it is necessary to employ feature selection methods in order to enhance the accuracy of the machine learning classifiers [13].

In order to enhance the effectiveness of widely used machine learning algorithms, many feature selection techniques have been employed to identify the most important features in malignant microarray datasets [14,15,16,17,18]. Even though filter feature selection approaches offer computational efficiency and the ability to reduce the dimensionality of microarray datasets, their accuracy results are limited since they evaluate features independently of classifiers. On the other hand, wrapper feature selection approaches interact with the classifier throughout the feature evaluation process, resulting in superior outcomes compared to the filter method. Nevertheless, the utilization of wrapper approaches on high-dimensional microarray datasets might be difficult and time-consuming.

In recent years, several evolutionary and bio-inspired algorithms [19,20,21,22,23,24,25,26,27,28,29,30,31,32,33,34,35] have been implemented in literature to obtain the highest level of accuracy in the gene selection challenge. Although feature selection methods based on evolutionary algorithms can overcome the limitations of filter and wrapper methods, they may result in greater computational times for certain machine learning algorithms. Due to the high dimensionality and large number of features in malignant microarray datasets, it is not feasible to initially employ evolutionary algorithms as feature selection approaches. It is essential to reduce the features of microarray cancer datasets using filter feature selection. Then, an evolutionary optimization algorithm can be utilized to optimize the features further to maximize cancer classification performance. This motivated us to suggest a novel hybrid filter-differential evolutionary feature selection method that combines the strengths of both filters and evolutionary techniques to generate effective solutions with improved cancer classification performance for high-dimensional microarray datasets.

The Differential Evolutionary (DE) is one of the superior optimization evolutionary algorithms, which is inspired by the biological evolution of the chromosomes in nature. DE performs well in convergence, although it is straightforward to implement and requires a few parameters to control and low space complexity. These attractive advantages of DE over other competitive optimization algorithms make DE gained widespread recognition for its exceptional efficacy in addressing various optimization challenges. Hence, this study aims to combine the superior performance of the DE optimization algorithm with filter selection methods to improve the classification accuracy of four microarray datasets by highlighting the most important and relevant genes. This is the first attempt at applying the hybrid filter and DE-based gene selection and classification of DNA Microarray data to the belief of our knowledge. In this paper, we propose a novel approach that combines feature selection methods based on differential evolutionary optimization algorithms and filter methods for identifying the most effective subset of features. Six common filtering methods were applied in this study to assign a score to each feature in microarray cancer datasets. These methods were then used to reduce the dimensionality of the datasets by retaining only the highest-ranked features and removing superfluous and irrelevant ones. The DE algorithm was then used to optimize the reduced cancer datasets, resulting in significantly improved results in cancer classification. Our proposed approach improved the classification performance of cancer when applied to microarray datasets with high dimensions. The following is a summary of the most important contributions that our paper represents:Feature reduction using filter algorithms: Even though microarray datasets are used a lot in scientific literature, a recent study [28] found that popular machine learning methods weren’t very good at correctly classifying the high-dimensional microarray datasets such as the Brain, Breast, Lung, and CNS datasets since they have limited samples and thousands of redundant, irrelevant, and noisy genes or features. In this paper, we employed six well-known, fast and effective filtering methods—Information gain (IG), information gain ratio (IGR), correlation (CR), Gini index (GIND), Relief (RELIEF) and Chi-squared (CHSQR) to reduce the dimensionality of microarray datasets by ranking the features and genes and then selecting only the top 5% of ranked features in order to enhance the cancer classification performance.Optimal feature selection using DE: The performance improvement of the machine learning algorithms achieved on high-dimensional microarray datasets considering only the filter methods is inadequate since the filter methods assess the features separately by identifying the correlation between each feature and the class label. Therefore, the DE optimization algorithm is suggested to be effectively used in the second phase to identify the correlation between a set of the best features and the class label. This is done to further optimize the selected features obtained from the filter methods and to make an enhancement in cancer classification performance. DE is able to reduce almost 50% of the irrelevant features in comparison to filter feature selection methods.Maximizing the cancer classification: The proposed hybrid filter-DE feature selection methods have achieved the two main objectives stated in this study, which are reducing the dimensionality and increasing the cancer classification performance on the four microarray datasets used. It is evident from experimental results that proposed hybrid filter-DE feature selection methods accomplished excellent performances with fewer important features: 100% classification accuracy with only 121 features for the Brain dataset, 100% classification accuracy with only 156 features for CNS, 98% classification accuracy with only 296 features for Lung dataset, and 93% classification accuracy with only 615 features for Breast, respectively. The average improvement percentages of accuracy achieved by the proposed methods were up to 42.47%, 57.45%, 16.28% and 43.57% compared to the previous works that are 41.43%, 53.66%, 17.53%, 61.70% on Brain, CNS, Lung and Breast datasets, respectively. Compared to the previous works, the proposed methods accomplished better improvement percentages on Brain and CNS datasets, comparable improvement percentages on Lung dataset, and less improvement percentages on Breast dataset.

The remaining part of this paper consists of the following sections. Section 2 discussed recent works related to cancerous gene selection and classification performance suggested for high-dimensional microarrays. Section 3 describes proposed methodology and elaborates on the details of the phases of the proposed hybrid filter-DE feature selection methods. The experimental results and discussion of the proposed hybrid filter-DE are presented in Section 4. Finally, the paper is concluded, and future work is recommended in Section 5.

## 2. Related Works

This section presents and investigates the existing hybrid feature selection methods approaches that have recently been applied on cancer microarray datasets to improve cancer classification results.

Karthika et al. [20] employed the mixture model (MM) in addition to the Fast Fourier Transform (FFT) on Microarray Gene Expression data for dimensionality reduction. In order to select an effective feature, they employed optimization techniques called Dragonfly. Nonlinear Regression, DT, RF, and SVM were used as classifiers in this study. The classifiers’ performance is evaluated both with and without feature selection methods. Finally, hyper-parameter tuning techniques such as Adaptive Moment Estimation (Adam) and Random Adaptive Moment Estimation (RanAdam) are used to improve classifiers, resulting in an accuracy of approximately 98% with the SVM classifier. This research did not address computational complexity and model validation techniques. In addition, this study has notable limitations, including population-specific findings, reliance on MAGE data, and the influence of outliers.

Elbashir et al. [21]. suggested a graph attention network (GAT) model to utilize diverse mRNA and miRNA for the prediction of the survival rate of non-small cell lung cancer (NSCLC) using multi-omics data. Chi-square analysis was used to select the most significant features to include in the model. They used the synthetic minority oversampling method (SMOTE) to make the dataset and the concordance index (C-index) more equal, and they tested the model on different sets of omics data. When using combined mRNA and miRNA data, they obtained the highest value of the C-index (0.82) along with the accuracy of 0.75. Chi-Square approaches cannot be regarded as the most ideal feature selection method for highly complicated and correlated biological data; this is a significant limitation of the current research.

Zamri et al. [22] presented a hybrid metaheuristics optimization-based two-stage feature selection model. The SKF-MUT simulated Kalman filter was used in this study to pick microarray features that would make the ANN classification more accurate. The experimental results were validated using eight binary and multiclass benchmark datasets. SKF-MUT effectively selected the correct number of features and achieved 95–100% classification accuracy. The significant limitations of this study include model evaluation relying just on accuracy. Instead, other metrics like precision, recall, F1-score, or AUC-ROC might better assess the model’s performance along with accuracy. Further, the computational cost of feature selection has not been discussed.

Ali et al. [23] presented a hybrid filter-genetic feature selection method to reduce microarray dataset dimensionality. The first part of this work used three filter methods: information gain (IG), information gain ratio (IGR), and Chi-squared (CS) to pick the most relevant microarray dataset features. The second phase used a genetic algorithm to optimize the features selected in the first phase of the proposed approach. The proposed method was validated utilizing breast, lung, CNS, and brain cancer microarray datasets. Experimental results indicated the suggested model improved performance of various common machine learning approaches in terms of Accuracy, Recall, Precision, and F-measure and the reported accuracy ranges from 92 to 100%. The limitations of the existing work can be included as computational cost of the feature selection process and also statistical validation not discussed.

Elemam and Elshrkawey [24] introduced a two-stage hybrid feature selection. They began by using feature evaluation methods that included chi-squared, F-statistics, and mutual information (MI) filters. In the second phase, they employed wrapper-based sequential forward selection with ML models like SVM, DT, RF, and KNN classifiers to find the optimal set of features. The model was then rigorously tested and validated using lung cancer, ovarian cancer, leukemia, and SRBCT datasets. The results were impressive, with an accuracy rate of almost 100 percent and a minimal number of selected features. However, the study’s performance was solely measured through accuracy, and the issue of feature redundancy was not adequately addressed. No statistical tests were conducted for model validation, which are the limitations of the existing work

In a recent study, Abasabadi et al. [25] proposed a novel hybrid feature selection method to address the challenge of high dimensionality in microarray datasets. The methodology combines a filter approach (SLI-γ) with a genetic algorithm (GA). In the initial phase, 99% of irrelevant features were eliminated using SLI-γ. The second phase involved the GA optimization of the remaining relevant features to enhance classification accuracy. The results of this method were not only enhanced performance but also a significant reduction in execution time, which is a remarkable achievement. However, the inherent computational complexity associated with GA-based optimization remains a challenge, especially as the dimensionality of datasets increases.

Almutiri et al. [26] proposed a hybrid feature selection method, GI-SVM-RFE, to improve classification accuracy in high-dimensional microarray datasets. The methodology combines the Gini index and SVM-RFE to select informative genes recursively. The results showed enhanced classification accuracy reported as 90.67 compared to other methods without feature selection or using only the Gini index or SVM-RFE. The model not validated statistically.

Similarly, Xie et al. [27] proposed the Multi-Fitness RankAggreg Genetic Algorithm (MFRAG). The methodology employed a genetic algorithm framework to integrate nine feature selection techniques. It uses an ensemble model to assess fitness and guide the evolutionary process. The results indicated that MFRAG demonstrated exceptional performance, achieving an accuracy between 87 and 100 percent, with increased classification accuracy using fewer selected characteristics. The limitations of this study include the potential for overfitting despite the use of the ensemble method and the absence of statistical discussion of model validation.

Dash et al. [28] proposed a hybrid methodology for feature reduction utilizing harmony search and Pareto optimization. The authors employed the Harmony Search algorithm and Gene Selection (AHSGS) to identify the top 100 gene characteristics while also utilizing Bi-objective Pareto optimization to eliminate insignificant gene features. The model was assessed using four publicly available microarray datasets. In all instances, SVM surpassed other classifiers, attaining nearly 100 percent accuracy, with the exception of the Colon dataset, where ANN reached 82 percent accuracy. The existing work exhibits notable limitations, particularly in the statistical analysis, as the results concerning significance levels are absent. Furthermore, the author failed to address the criteria or methodology employed to ascertain the Harmony Memory Consideration Rate and Pitch Adjusting Rate.

Almutiri et al. [29] suggested a fusion-based feature selection framework aimed at mitigating high dimensionality and enhancing classification performance in gene expression microarray data. The framework utilizes a three-layer approach. The first layer has independent feature selection methods for gene ranking and scoring. The second layer consists of a threshold-based filtering step and a final decision layer employing majority or consensus voting. Experiments were conducted on five microarray datasets using an SVM classifier. The results revealed enhanced classification accuracy, achieving up to 97% on the Prostate dataset, alongside dimensionality reduction in comparison to existing methods. The primary limitations of this study are threshold sensitivity and dependence on voting strategy.

Kilicarslan et al. [30] proposed a hybrid model to significantly improve cancer diagnosis. The methodology combined relief and stacked autoencoders for dimension reduction. Then, SVM and CNN were used to improve classification accuracy. The proposed method achieved the highest classification accuracies (98.6%, 99.86%, and 83.95%, respectively) on three microarray datasets (Ovarian, Leukemia, and CNS), outperforming SVM and other tested approaches. The study highlighted the effectiveness of dimension reduction in enhancing classification accuracy. However, this study has most notable limitations as limited comparison with other Feature Selection Methods and process of hyperparameter optimization for the CNN model.

Baliarsingh et al. [31] proposed a microarray-based hybrid cancer classification model. The methodology utilized ANOVA to select relevant genes. Then, the enhanced Jaya (EJaya) algorithm and the forest optimization algorithm (FOA) were utilized to find the best gene subset, and SVM was used for classification. The proposed method reduced features and exceeded benchmark methods in classification accuracy from 96 to 100%. The significant limitation of this study is the use of a single classifier (SVM), which may not generalize across datasets. In addition Parameter tuning may also affect EJaya and FOA algorithm performance.

In this study, Almugren and Alshamlan [32] evaluated and compared contemporary hybrid approaches combining bio-inspired evolutionary algorithms for gene selection and cancer classification. The methodology, which was conducted with utmost thoroughness, involved reviewing various algorithms, with a focus on genetic algorithms (GA) as wrapper methods for gene selection. The results revealed that GA is the most extensively used and achieved the highest accuracy with a minimal number of selected genes ranging from 93 to 100%. In contrast, the Firefly algorithm has not been used as a wrapper approach. The limitation of existing work is the inadequate investigation of alternative hybrid algorithms.

Sayed et al. [33] proposed this study to investigate the efficacy of a Nested Genetic Algorithm (Nested-GA) for feature selection in high-dimensional colon cancer microarray datasets. The methodology used a *t*-test to preprocess data and a nested approach with two Genetic Algorithms. The outer Genetic Algorithm (OGA-SVM) is used for gene expression data, and the Inner Genetic Algorithm (IGA-NNW) is utilized for DNA methylation data. The validation was performed using five cross-folds, ensuring a thorough examination of the results. Nested-GA outperformed KNN and RF on the colon cancer dataset with 99.9% classification accuracy. This study’s main limitation is comparing Nested-GA to a limited set of feature selection algorithms (KNN and RF). A more extensive comparison with additional contemporary methods could yield more significant insights into its performance which is one of the limitation of this work.

Similarly, Ghosh et al. [34] introduced a novel two-stage hybrid model that integrates multiple filter methods with a genetic algorithm (GA) for cancer detection in microarray datasets. The methodology involved initially creating an ensemble of filter methods such as ReliefF, chi-square, and symmetrical uncertainty by looking at the union and intersection of their top-n-ranked features. Then, in the next step, GA is used to make the results of the first step even better. The result showed that the model did better than the best current methods, with an accuracy of about 100% and a smaller number of chosen features across five cancer datasets: colon, lung, leukemia, SRBCT, and prostate. The limitation of this study is that the performance evaluation is mainly based on accuracy and feature count.

Hameed et al. [35] introduced a three-phase hybrid method to select and classify high-dimensional microarray data. To achieve this purpose, the author employed Pearson’s Correlation Coefficient (PCC) alongside Binary Particle Swarm Optimization (BPSO) or Genetic Algorithm (GA) and numerous classifiers. In the first phase, the methodology utilizes PCC as a filter for feature selection. Subsequently, the second phase involved the application of either BPSO or GA as wrapper methods. The data was classified using five distinct classifiers. The results showed improved classification accuracy, with BPSO outperforming GA in speed and effectiveness across multiple datasets and classifiers. Although the authors compared BPSO with GA, they did not study a broader range of optimization algorithms or hybrid approaches. This highlights the urgent need for a more comprehensive understanding of the best practices in feature selection and classification.

As can be observed from the existing works discussed above, the filter methods have been utilized individually [18,24] or combined with the genetic algorithm [23,24,26,27,33,34,35] or wrapper feature selection [29,32] in order to improve cancer classification on microarray datasets. In contrast, this study proposed integrating Differential Evolution (DE) with some popular filter methods to maximize cancer classification on microarray datasets. DE has several attractive advantages over other competitive optimization algorithms. DE performs well in convergence, although it is straightforward to implement and requires a few parameters to control and low space complexity. Therefore, the proposed hybrid method successfully identified the most effective features using both filter methods and DE. This resulted in improved performance of classifying cancer on highly dimensional microarray datasets.

## 3. Materials and Methods

This section introduces a hybrid feature selection strategy that combines filter methods with differential evolutionary algorithms. This approach aims to effectively identify the most relevant features from high-dimensional microarray datasets to get excellent results in cancer classification. The methodology of the proposed hybrid filter and DE-based feature selection consists of five phases, as shown in Figure 1: microarray data collection, feature reduction using filter algorithms, feature selection using DE, training, and testing and evaluation of trained models.

### 3.1. Description of Cancerous Microarray Data Used

There are many microarray datasets used in literature. In order to assess the effectiveness of the suggested hybrid filter-DE feature selection method, we will focus on four malignant microarray datasets. Brain cancer [36,37], Breast cancer [38], Lung cancer [39] and Central Nervous System (CNS) [39] datasets since these datasets are more common microarray datasets and have high-dimensional features. Furthermore, several latest works [20,21,22,23,24,25,26,27,28,29,30,31,32,33,34,35] reported that the high-dimensional features of these four cancerous microarray datasets caused low classification results. Table 1. shows characteristics of the four cancerous microarray datasets used for assessing the proposed hybrid filter-DE feature selection method.

### 3.2. Feature Reduction Using Filter Algorithms

As stated in Section 3.1, the microarray datasets included in this study are characterized by a high number of dimensions and contain numerous duplicated and irrelevant features. It is impractical to train machine learning algorithms with all these features of microarray datasets. So, we use common filter feature selection methods to rank all features of microarray cancer datasets. Subsequently, to reduce the high dimensional datasets, only the top 5% of the best-ranked features are selected, and the remaining redundant and unnecessary features are removed. In this work, the features of microarray cancer datasets are ranked with various common filter feature selection methods. The features of microarray cancer datasets are ranked by utilizing well-known, fast and effective filtering methods such as correlation, information gain, and information gain ratio, Relief, Chi-squared and Gini Index.

#### 3.2.1. Correlation

The correlation-based feature selection (CR-FS) technique is a widely used filter algorithm [40] that relies on the correlation between features and the target class. CFS chooses only the features that have high correlation to the target class and minimal correlation with each other. CFS uses the following Equation (1) to evaluate the features.
(1)$$F_{s}=\frac{f\bar{v_{ca}}}{\sqrt{f+f\left(f-1\right)\bar{v_{aa}}}}$$
where the average correlation between the class and the feature is represented by $\bar{v_{ca}}$ while the correlation between two features is represented by $\bar{v_{aa}}$, and f denotes the number of features. In this study, the correlations between features and the target class are computed by feature Pearson’s correlation coefficient (PCC).

#### 3.2.2. Information Gain

The information gain (IG) [41] is a well-known filter technique that has been effectively used to choose highly relevant features by employing the entropy concept to assess the importance of features. In IG, the worth of an attribute is determined by computing the quantity of information gained by the feature with respect to the target class. The IG method uses Equation (2) to compute the score IG (S, A) of a feature A:(2)$$IG\left(S,A\right)=Entropy\left(S\right)-\sum_{v\in Value\left(A\right)}\frac{\left|S_{v}\right|}{\left|S\right|}Entropy\left(S_{v}\right)$$
where $S_{v}$ represents the subset of data S and feature A has the specific value v.

By using Equation (3), Entropy(S) is calculated based on the probability $P(c_{j})$ of class $c_{j}$ in S.
(3)$$Entropy(S)=-\sum_{j=1}^{c}P(c_{j})\log_{2}P(c_{j})$$

#### 3.2.3. Information Gain Ratio

Although IG is usually a good filter feature selection method used for ranking the features, IG biases towards features with a large number of distinct values. The information gain ratio (IGR) [42], which penalizes features with a large number of values, is used to address the drawback of IG. In IGR, the information gain is divided by the split information, which effectively measures the inherent information required to distinguish between the various values of that feature. IGR uses an Equation (4) to calculate split information (S, A).
(4)$$\mathrm{Split}\;\mathrm{information}\left(S,A\right)=-\sum_{i=1}^{m}\frac{\left|S_{i}\right|}{\left|S\right|}\log_{2}\frac{\left|S_{i}\right|}{\left|S\right|}$$

The original dataset is denoted as S, each sub-dataset after being split is represented as $S_{i}$, and m is the number of sub-datasets. The number of samples in S and $S_{i}$ are denoted as $\left|S\right|$ and $\left|S_{i}\right|$ respectively.

The score IGR (S, A) of feature A is calculated by using Equation (5).
(5)$$IGR\left(S,A\right)=\frac{IG\left(S,A\right)}{Split\;information(S,A)}$$

#### 3.2.4. Relief

The Relief (RF) feature selection method [43] is another popular filter method based on the nearest neighbor to weight attributes, so it is utilized to deal effectively with dependent features and noisy data. The Relief feature selection method arbitrarily chooses a sample of the data and then finds its k-nearest neighbors from the same class and each of the opposite classes. The Relief method uses Equation (6) to compute the score $S_{i}\;$ of ith attribute:(6)$$S_{i}=\frac{1}{2}\sum_{b=1}^{l}d\left(X_{ib}-X_{{iM}_{b}}\right)-d\left(X_{ib}-X_{{iH}_{b}}\right)$$
where l represents the randomly chosen samples from the dataset, $d\left(X_{ib}-X_{{iM}_{b}}\right)$ denotes the distance between the ith attribute value of a randomly chosen sample $X_{ib}$ and the nearest sample $X_{{iM}_{b}}$ of the same class, while and $d(X_{ib}-X_{{iH}_{b}})$ represents the distance between the ith attribute value of a randomly chosen sample $X_{ib}$ and the nearest sample $X_{{iH}_{b}}$ of the different class.

#### 3.2.5. Chi-Squared

Chi-squared [44] is a popular and statistical filter method based on calculating the dependence between features and class. Features are ranked based on how strongly they are associated with the target class. The Chi-squared (CHSQR) test computes the aggregate of squared differences between the observed frequencies of each category with the expected frequencies under the assumption of no association. By computing χ^2^ with regard to the class as shown in Equation (7), each feature’s significance is assessed.
(7)$$X^{2}=\sum_{i=1}^{r}\sum_{j=1}^{c}\frac{{\left(O_{ij}-E_{ij}\right)}^{2}}{E_{ij}}$$
where c represents the class number and r stands for the number of bins utilized to discretize numerical attributes. $O_{ij}$ and $E_{ij}$ stand for the observed frequency and expected frequency, respectively.

#### 3.2.6. Gini Index

Another popular filter method used for feature ranking is the Gini index (GI) [45]. It computes and allocates a weight or scoring to each feature, indicating the feature’s ability to distinguish instances from distinct classes. The following Equation (8) is utilized for calculating the Gini index of S:(8)$$Gini\left(S\right)=1-\sum_{j=1}^{c}{P(c_{j})}^{2}$$
where c denotes number of classes and $P\left(c_{j}\right)$ refers to the probability of samples belonging to class $c_{j}$ in S.

### 3.3. Differential Evolution Based Feature Selection

Machine learning algorithms can be effectively trained to produce enhanced classification results using reduced cancer datasets that only include the top-ranked features determined by the filter methods. However, the performance of the machine learning algorithms considering only the filter methods is still limited when they are applied on high-dimensional microarray datasets [46]. This is because the majority of filter methods ignore the correlation between groups of features and the class label. They assume that the features are independent and find the relationship between the individual feature and the class label. Furthermore, filter methods use certain criteria to assess the features without the use of a machine learning algorithm. Thus, in order to improve the effectiveness of the cancer classification, Differential Evolution (DE) is used in this paper to further optimize the chosen features that are selected by the filter approaches.

DE is a powerful global optimization approach developed by Storn and Price [47], which belongs to category of evolutionary algorithms that draw inspiration from the natural evolution of chromosomes. In recent years, DE has been applied successfully in many real applications and optimization problems since it has several attractive advantages [46,48] over other competitive optimization algorithms. DE performs well in convergence although it is straightforward to implement and requires a few parameters to control and low space complexity.

Firstly, the initial population of the possible solutions in DE is randomly generated over the feasible region. Accordingly, the fitness function is used to evaluate the candidate solutions of the initial population. Then, DE generates a new solution by combining several solutions with the candidate solution. Like GA, the candidate solutions in DE population iteratively evolve to find better solutions through repeated generations of three main DE operators: mutation, crossover, and selection. However, the mutation, crossover, and selection operations in DE are conducted in different ways compared to GA operations.

Let $X_{i}=\left(x_{i1},x_{i2},x_{i3},\ldots ,x_{id},x_{iD}\right)$ be the source vector of solution i, where i = {1, 2, …, P} and P is population size, d = {1, 2, …, D}, and D is the dimensionality of the search space. The mutation, crossover, and selection in DE are achieved as follows:Mutation: For each solution vector $X_{i}$, Equation (9) is used to produce a mutant solution $V_{i}$
(9)$$V_{i}=X_{r_{1}}+F\left(X_{r_{3}}-X_{r_{2}}\right)$$
where F is the mutation factor in the interval [0, 1], $X_{r_{1}}$, $X_{r_{2}}$, $X_{r_{3}}$ are three individuals or candidate solutions which are randomly chosen from the population such that r1≠r2≠r3≠i.

Crossover: The crossover operation is performed between the parent $X_{i}$ and its corresponding mutant solution $V_{i}$ in order to produce a trial solution $U_{i}=\left\{u_{i1},u_{i2},u_{i3},\ldots ,u_{id},u_{iD}\right\}$ as shown in Equation (10).(10)$$u_{id}=\left\{\begin{array}{l} v_{id},\;\;\;\;\mathrm{if}\;\delta \leq \mathrm{CR}\;\mathrm{or}\;d=d_{rand}\\ x_{id},\;\;\;\;\mathrm{otherwise} \end{array}\right.$$
where CR represents crossover rate which is user-predefined constant within the range of [0, 1], δ is a random number between [0, 1], and $d_{rand}$ is randomly selected index ∈[1, D].

Selection: The fitness of trial vector $U_{i}$ is computed and then compared with the fitness of source vector $X_{i}$. If $U_{i}$ i is better than $X_{i}$, $X_{i}$ will be replaced with $U_{i}$ into the population of the next generation. Otherwise, the population keeps the source vector $X_{i}$ for the next generation. 

From generation to generation, the mutation, crossover and selection are applied to update and evolve the population in order to find the optimal solution until a stopping criterion is met.

DE was originally produced for solving optimization problems in continuous numbers search space. Therefore, the binary version of DE(BDE) was developed in order to solve feature selection and other binary optimization problems. The steps of feature selection based on DE can be described as follows:Initialization: In this step, the initial population of possible individuals is generated randomly. Each individual represents a possible solution of feature subset which is binary-coded vector with m bits, where m is the number of available features. If any bit of an individual is 1, the corresponding feature is selected; otherwise, it is not selected. For example, five features (3rd, 5th, 7th, 8th and 9th features) are selected for given a solution X = {0, 0, 1, 0, 1, 0, 1, 1, 1, 0}.Fitness evaluations: In this step, the individuals (feature subset solutions) in the population are evaluated using the fitness function. To calculate the fitness value of a feature subset solution, the misclassification rate (1-classification rate) of a machine learning technique trained with that feature subset solution is calculated for each individual in the population. That means the training dataset with the features indicated by 1 value on feature subset solution is used to train the machine learning technique. Then the misclassification rate is computed as the fitness of that feature subset solution. The misclassification rate is the ratio of the number of incorrectly classified instances to the total number of instances. The fitness function in the proposed DE-based feature selection aims at minimizing the misclassification rate of machine learning techniques trained with the possible feature subset solutions in order to identify the optimal feature subset that can produce the best classification results.Mutation process: For the current individual $X_{i}$, the mutation process in DE starts by randomly choosing three individuals $X_{r_{1}}$, $X_{r_{2}}$, $X_{r_{3}}$ from the population where r1≠r2≠r3≠i. In DE used in feature selection problem, mutant solution $V_{i}$ is generated based on difference vector as shown in the following Equations (11) and (12).


(11)
$${difference\;vector}_{id}=\left\{\begin{array}{l} \;\;\;\;\;0,\;\;\;\;\mathrm{if}\;x_{r_{1}d}=x_{r_{2}d}\\ x_{r_{1}d},\;\;\;\;\mathrm{otherwise} \end{array}\right.$$




(12)
$$v_{id}=\left\{\begin{array}{l} \;\;\;\;\;1,\;\;\;\;\mathrm{if}\;{difference\;vector}_{id}=1\\ x_{r_{3}d},\;\;\;\;\mathrm{otherwise} \end{array}\right.$$



Crossover process: Once the mutant individual is generated in the mutation process, the trial individual $U_{i}=\left\{u_{i1},u_{i2},u_{i3},\ldots ,u_{id},u_{iD}\right\}$ is created by performing the crossover process between the source vector $X_{i}$ and its corresponding mutant vector $V_{i}$. The trial solution $U_{i}$ in DE-based feature selection is computed using the same equation used in the original DE as shown in Equation (13).

(13)$$u_{id}=\left\{\begin{array}{l} v_{id},\;\;\;\;\mathrm{if}\;\delta \leq \mathrm{CR}\;\mathrm{or}\;d=d_{rand}\\ x_{id},\;\;\;\;\mathrm{otherwise} \end{array}\right.$$
where CR represents crossover rate which is user-predefined constant within the range of [0, 1], δ is a random number between [0, 1], d = {1, 2, …, D}, and D is the dimensionality of the search space, and $d_{rand}$ is randomly selected index ∈[1, D].

Selection process: In this step, DE compares the fitness values (misclassification rate) produced by the source solution $X_{i}$ and the trial solution $U_{i}$. If the trial solution $U_{i}$ has better fitness value (lower misclassification rate) than the source solution $X_{i}$, DE replaces the source solution $X_{i}$ with the trial solution $U_{i}$ in the population for the next generation. Otherwise, the source solution $X_{i}$ is retained in the DE population for the next generation.

In DE evolutionary process, the mutation, crossover, and selection processes are repeatedly conducted until a stopping criterion is satisfied. Eventually, the DE returns the best solution in the DE population that represents the optimum feature subset that can be effectively utilized later in the training and testing phases. Algorithm 1 presents steps of the proposed hybrid filter and differential evolution-based feature selection suggested to enhance cancer classification in Microarray data.
**Algorithm 1:** Hybrid filter and differential evolution-based feature selection
**Input:** F: Original feature set, and P: Size of population**Output:** SF: The optimal selected features
**Begin**1Compute scores of the features using Filter methods: Information gain (IG), information gain ratio (IGR), correlation (CR), Gini index (GIND), Relief (RELIEF) and Chi-squared (CHSQR)2Do ranking of all features in F based on the scores computed by Filter methods3Reduce dimension of training data by selecting only the top 5% of ranked features4Set Crossover rate, maximum number of generations Max_t, the generation counter t = 05Generate and initialize P individuals of population (feature subset solutions)6**While** t < Max_t **and** Stopping criterion is not satisfied **Do**7 t = t + 18 Compute the fitness of individuals using misclassification rate9 Best individual = Evaluate the fitness of individuals10 **For** each individual $X_{i}$ **Do**
11  Choose three individuals $X_{r_{1}}$,
$X_{r_{2}}$,
$X_{r_{3}}$ randomly   where
r1≠r2≠r3≠i
12  Generate a mutant solution $V_{i}$ using Equations (9) and (10)13  Generate a trial vector $U_{i}$ using Equation (11)14  Evaluate fitness values of $X_{i}$ and
$U_{i}$
15  **If** fitness of
$U_{i}$ is better than fitness of
$X_{i}$
16        $X_{i}$ is replaced with $U_{i}$ in the population and then         $U_{i}$ used for the next generation17  
**Else**
18     $X_{i}$ is kept for next generation19  
**End if**
20 
**End For**
21**End While**22Extract the optimal selected features from the individual with the best fitness value23Return the optimal features SF24**End Algorithm**

### 3.4. Training of Machine Learning Techniques

As mentioned, the high-dimensional microarray datasets are reduced by applying the proposed hybrid filter-DE feature selection. In the first stage, we used filter methods to rank all features in microarray datasets and then select only the top 5% of features while the remaining features were removed. Then, we used DE in the second stage of the proposed method to choose only highly relevant features of the reduced microarray datasets. By using the final training microarray datasets with the most optimal features, we trained some well-known machine learning algorithms, such as support vector machine (SVM), naïve Bayes classifier (NB), k-Nearest Neighbour (kNN), decision tree (DT), and random forest (RF), that are widely employed in the literature to classify cancer having large dimension microarray datasets. To optimally design the kNN, NB, DT, RF, and SVM, the best features selected by the proposed hybrid filter-DE feature selection are used to as inputs to train these models. In addition, the best settings and parameters used in all classifiers were selected by a trial-and-error basis in order to produce the best results. Furthermore, we train these models using stratified cross-validation that is especially useful for imbalanced datasets, ensuring equal representation in each fold.

### 3.5. Testing and Evaluation of Machine Learning Techniques

After completing the training phase, the trained classification models based on the proposed hybrid filter-DE feature selection method will be evaluated with the new testing microarray datasets. Instead of using all the features of testing microarray datasets, we reduce testing microarray datasets by using only the optimal features selected by the proposed hybrid filter-DE feature selection method. Then, the reduced testing microarray datasets will be used as input of SVM, NB, kNN, DT, and RF to assess the cancer classification performance of these classifiers. In this paper, we used 10-fold cross-validation to evaluate the proposed method. We use some popular evaluation metrics used in the literature to evaluate the proposed hybrid filter-DE feature selection method, such as classification accuracy, recall, precision, and F-measure. The evaluation metrics used in this paper are defined based on the confusion matrix as follows:

Classification accuracy expressed in Equation (14) measures the total proportion of properly diagnosed samples.
(14)$$Accuracy=\frac{TP+TN}{TP+FP+FN+TN}\times 100\left(\%\right)$$

Equation (15) calculates Recall measure, which is the proportion of positive samples that are properly diagnosed as the positive class.
(15)$$Recall=\frac{TP}{TP+FN}\times 100\left(\%\right)$$

Precision is calculated in Equation (16) as the number of properly diagnosed positive samples divided by the total number of samples classified as positive. It is represented by the following formula:(16)$$Precision=\frac{TP}{TP+FP}\times 100\left(\%\right)$$

F-measure is a measure that combines Recall and Precision into a harmonic mean as expressed in Equation (17) to give a balanced assessment of the model’s performance.
(17)$$F-measure=2\times \frac{Precision\times Recall}{Precision+Recall}\times 100\left(\%\right)$$
where TP denotes the number of correctly classified positive samples, FP stands for the number of negative samples incorrectly classified as positive samples, TN stands for the number of correctly classified negative samples, and FN indicates the number of positive samples incorrectly classified as negative samples.

In the multi-class classification, these measures are computed using Equations (18)–(23). Recall for each class *i* is calculated using Equation (18), which represents the ratio of true positive predictions to the total actual samples of that class. The Overall Recall can be calculated using Equation (19).
(18)$${Recall}_{i}=\frac{TP_{i}}{TP_{i}+FN_{i}}\times 100\left(\%\right)$$
(19)$$Overall\;Recall=(\frac{1}{n}\sum_{i=1}^{n}{Recall}_{i})\times 100\left(\%\right)$$
where ${TP}_{i}$, ${FP}_{i}$, ${TN}_{i}$ and ${FN}_{i}$ are true positive, false positive, true negative and false negative for class i, respectively, and n is the number of classes.

As shown in Equation (20), Precision for each class *i* is the ratio of true positive predictions for that class to the total samples classified as that class. The Overall Precision can be calculated using Equation (21).
(20)$${Precision}_{i}=\frac{TP_{i}}{TP_{i}+FP_{i}}\times 100\left(\%\right)$$
(21)$$Overall\;Precision=(\frac{1}{n}\sum_{i=1}^{n}{Precision}_{i})\times 100\left(\%\right)$$

F-measure for each class *i* is calculated using Equation (22), which is the harmonic mean of precision and recall for that class. The Overall F-measure can be calculated using Equation (23).
(22)$${F-measure}_{i}=2\times \frac{{Precision}_{i}\times {Recall}_{i}}{{Precision}_{i}+{Recall}_{i}}\times 100\left(\%\right)$$
(23)$$Overall\;F-measure=(\frac{1}{n}\sum_{i=1}^{n}{F-measure}_{i})\times 100\left(\%\right)$$

In addition to classification measures, the classification error can be measured for each class using False Positive Rate and False Negative Rate. False Positive Rate (FPR) for each class *i* is calculated using Equation (24), which represents the ratio of samples incorrectly classified as class *i* out of all samples that do not actually belong to class *i*. Equation (25) is used to calculate the Overall FPR. In contrast, the False Negative Rate (FNR) for each class *i* is calculated using Equation (26), which denotes the proportion of samples that actually belong to class *i* but they are incorrectly classified as a different class. Equation (27) is used to calculate the overall FNR.
(24)$${FPR}_{i}=\frac{FP_{i}}{FP_{i}+TN_{i}}$$
(25)$$Overall\;FPR=\frac{1}{n}\sum_{i=1}^{n}{FPR}_{i}$$
(26)$${FNR}_{i}=\frac{FN_{i}}{FN_{i}+TP_{i}}$$
(27)$$Overall\;FNR=\frac{1}{n}\sum_{i=1}^{n}{FNR}_{i}$$

## 4. Experimental Environment and Results Discussion

This section presents the experimental environment and settings used to implement the suggested hybrid filter-DE techniques. In addition, it investigates the effectiveness of machine learning approaches after implementing the recommended hybrid filter-DE feature selection strategy, in comparison to their performance standalone and after applying only filter methods. The comparison includes the performance of machine learning by considering all features, features selected by filter methods, and features selected by the suggested hybrid filter-DE techniques. The most optimal classification results of the filter approaches were accomplished by training machine learning algorithms with top 5% of ranking features on these four datasets.

### 4.1. Experimental Environment

This study utilized the RapidMiner tool (version 10.1) and the Anaconda package. RapidMiner resources offer a comprehensive collection of machine learning algorithms and a variety of strategies for data validation. Furthermore, the Anaconda package is employed to execute differential evolutionary algorithms in the Python programming language, as well as for the purposes of training, testing, and visualization. As shown in Table 2, we have conducted many experiments and scenarios to get the best crossover rate (CR) and population size (P) that can produce the best classification results. The best parameters of DE used with the proposed hybrid filter-DE feature selection are listed in Table 3.

In addition, several Python libraries, such as Scikit-learn and Matplotlib, have been utilized. The data was visualized using Matplotlib, and machine learning techniques were implemented using the Scikit-learn module. The computing environment utilized a Lenovo Laptop equipped with the Windows 10 operating system, an Intel Core i7 (RTX) CPU, and 32 GB of RAM.

### 4.2. Results Analysis of Brain Dataset

It is evident from Figure 2 that when the IG filter was applied, the classification accuracies of KNN (78.57%), NB (69.05%), DT (50%), RF (78.57%), and SVM (69%) improved to 81%, 88.1%, 61.9%, 90.5%, and 81%, respectively. When IGR was used, the accuracy results of KNN, NB, DT, RF, and SVM showed enhancement and reached 83.33%, 85.71%, 64.29%, 90.86%, and 66.67%, respectively. Furthermore, when Chi-square (CHSQR) was applied, the accuracy results of KNN, NB, DT, RF, and SVM were enhanced to 80.95%, 83.33%, 69.05%, 88.1%, and 66.67%, respectively. In addition, when applying correlation CR, the accuracy results showed some improvement to 88.1%, 71.43%, 61.1%, 88.1%, and 83.33%, respectively. In the case of the GIND filter, accuracy enhancement was observed as KNN (83.33%), NB (83.33%), DT (64.29%), RF (90.46%), and SVM (80.95%). Additionally, the relief filter method showed enhancement with KNN (88.1%), NB (80.95%), DT (61.09%), RF (92.86%), and SVM (71.90%).

The hybrid IG-DE method improved the results of classification accuracy of KNN (78.57%), NB (69.05%), DT (50%), RF (78.57%), and SVM (69%) to 92%, 100%, 92%, and 92%, respectively, whereas the hybrid IGR-DE method improved them to 92%, 100%, 100%, 92%, and 92%. The suggested hybrid CHSQR-DE approach improved classification accuracy of KNN, NB, DT, RF, and SVM to 92%, 100%, 100%, 92%, and 92%, whereas hybrid CR-DE improved them to 85%, 100%, 85%, 85%, and 85%. Furthermore, the proposed hybrid GIND-DE method improved accuracy results to 92%, 100%, 92%, and 92%, while the hybrid RELIEF-DE method improved accuracy results to 92%, 100%, 100%, 92%, and 92% when compared to all features’ accuracy results.

In terms of accuracy, the IG-DE-NB, IGR-DE, with NB and DT, CHSQR-DE with NB and DT, CR-DE with NB, GIND-DE with NB and DT, and RELIEF-DE with NB and DT classifiers achieved optimal performance of 100% accuracy using the top 5% of ranked features on this dataset.

Table 4 shows the comparison of the number of selected genes and other measures of KNN, NB, DT, RF, and SVM by applying all features, filter feature selection, and hybrid-DE filter feature selection methods. It is obvious from Table 4 that the machine learning algorithms, with the implementation of the proposed hybrid filter-DE methods, accomplished much better performance in terms of accuracy, precision, recall, and F-measure compared to their performance with all features and their performances considering only filter methods. Outstandingly, the proposed hybrid methods IG-DE, IGR-DE, CHSQR-DE, CR-DE, GIND-DE, and RELIEF-DE with NB achieved optimal 100% performance. Furthermore, it is obvious from Table 4 that CHSQR-DE with DT has an edge over all other hybrid methods in terms of the features selected. The smallest number of selected features are highlighted in grey color. CHSQR-DE with DT performed excellently by selecting only 121 features out of 5597 features. It is evident from Table 4 that the filter feature selection strategy reduced the number of relevant features in the BRAIN datasets from 5597 to 280. Furthermore, the suggested hybrid IG-DE, IGR-DE, CHSQR-DE, CR-DE, GIND-DE, and RELIEF-DE techniques in the BRAIN dataset on average favored only 142, 126, 134, 132, 144, 138 features out of the 280 features.

### 4.3. Results Analysis of CNS Dataset

Figure 3 demonstrates that the classification accuracies of KNN (61.67%), NB (61.67%), DT (58.33%), and RF (53.33%) were enhanced by applying IG to 75%, 66.67%, 80%, and 80% while they were enhanced by applying IGR to 68.33%, 78.33%, 68.33%, and 80%, respectively. Further, in the case of CHSQR, the observed enhancement reached 75%, 70%, 61.67%, and 83.33%. In the case of CR (correlation), the change was noted as 76.67%, 55.5, and 77.33%, respectively. Further, when GIND was applied, the enhanced results reached 75%, 80%, 68.33%, and 83.33%. The relief filter also showed enhancement, reaching 71.67%, 73.33%, 65%, and 75%. It was also observed that there was no enhancement in all cases with SVM.

The hybrid IG-DE technique improved the classification accuracies of KNN, NB, DT, RF, and SVM to 94%, 100%, 94%, 89%, and 94% respectively. On the other hand, the hybrid CHSQR-DE method improved the accuracies to 83%, 100%, 89%, 83%, and 89%. The hybrid CR-DE technique improved the classification accuracies of KNN, NB, DT, RF, and SVM to 945, 100%, 94%, 83%, and 94%, respectively. On the other hand, the hybrid GIND-DE method improved the accuracies to 94%, 100%, 94%, 89%, and 89%. In addition, their performances were improved by implementing the suggested hybrid RELIEF-DE technique to achieve success rates of 83%, 100%, 94%, 89%, and 83% respectively.

The least enhancements observed with the proposed IGR-DE were 72%, 94%, 94%, 83%, and 78% in comparison to KNN (61.67), NB (61.67), DT (58.33), RF (53.33), and SVM (65). In terms of accuracy, the IG-DE with NB, CR-DE with DT, GIND-DE with NB, CHSQR-DE with NB, and RELIEF-DE with NB classifiers achieved optimal performance of 100% accuracy using the top 5% of ranked features on this dataset. To further analyze the efficacy of the suggested technique, the results of the proposed approaches were compared against feature selection methods as well as all features as presented in Table 5. Table 5 shows the comparison of and the number of selected genes and other measures of KNN, NB, DT, RF, and SVM by applying all features, filter feature selection, and hybrid-DE filter feature selection methods.

Table 5 clearly shows that the machine learning algorithms performed significantly better performance in terms of accuracy, precision, recall, and F-measure after applying the proposed hybrid filter-DE methods compared to their performance with all features and their performance with only filter methods. Outstandingly, the proposed IG-DE with NB, CR-DE with DT, GIND-DE with NB, CHSQR-DE with NB, and RELIEF-DE with DT achieved optimal 100% performance. Furthermore, it is obvious from Table 5 that IG-DE with NB has an edge over all other hybrid methods in terms of the features selected. IG-DE with NB performed excellently with selecting only 156 features out of a total of 6129 features. The smallest number of selected features are highlighted in grey color. In addition, Table 5 shows that the filter feature selection strategy reduced the number of relevant features in the CNS datasets from 6129 to 306. Further, the suggested hybrid IG-DE, IGR-DE, CHSQR-DE, CR-DE, GIND-DE, and RELIEF-DE techniques choose only 163, 171, 177, 180, 174 and 178 relevant features on average from the 306 filtered features.

### 4.4. Results Analysis of Lung Dataset

Figure 4 shows that applying GIND improved the classification accuracies of KNN (92.61%), NB (90.15%), DT (84.43%), and RF (83.74%) by 93.6%, 95.07%, 91.13%, and 93.60, respectively. Also, after applying the IG filter method, accuracy results were enhanced to 92.61%, 95%, 86.7%, and 93.6%, respectively.

Further, when CHSQR was applied, the accuracy results showed some improvements at 92.12%, 92.12%, 85.22, and 92.61%, respectively. In the case of the IGR filter, enhancement was recorded with NB (93.6%), DT (86.21%), and RF (91.13%), while KNN and Relief showed a little decrement. In addition, with the CR filter method, the enhancement was recorded only with NB (94.09%) and RF (86.7%), while we observed a little decrease with KNN and DT. The relief filter method showed enhancement with NB (91.63%), DT (89.16%), and RF (92.12%) but it did not perform well with KNN, and a small amount of decrement was observed in the accuracy. No method of the six filter methods performed well with SVM, and no enhancement was recorded in accuracy.

The proposed hybrid IG-DE method enhanced further the classification accuracies of KNN, NB, DT, RF, and SVM to 97%, 97%, 97%, 93%, and 98%, respectively, while they were enhanced by applying the hybrid IGR-DE method to 95%, 92%, 98%, 95%, and 97%, respectively. The CHSQR-DE method enhanced further the classification accuracies of DT, RF, and SVM to 97%, 93%, and 97%, respectively, while for KNN and NB, a very small amount of decrement was observed. Additionally, by applying the hybrid CR-DE method, KNN, NB, DT, RF, and SVM results were enhanced to 95%, 93%, 93%, 93%, and 97%, respectively. Furthermore, by applying the hybrid GIND-DE method, the accuracy results of KNN, NB, DT, RF, and SVM were enhanced to 93%, 93%, 98%, 95%, and 95%, respectively, while they were enhanced by applying the hybrid RELIEF-DE method to 93%, 92%, 97%, 93%, and 93%, respectively. In terms of accuracy, the IG-DE with SVM, IGR-DE with DT, and GIND-DE with DT classifiers achieved optimal performance of 98% accuracy using the top 5% of ranked features on this Lung dataset.

Table 6 compares the performance measures, and the number of genes identified by the proposed methodology on the Lung dataset to five classifiers (KNN, NB, DT, RF, and SVM) with all features, filter feature selection, and hybrid-DE filter feature selection approaches. Table 6 clearly shows that the machine learning algorithms performed significantly better in terms of accuracy, precision, recall, and F-measure after applying the proposed hybrid filter-DE methods in comparison to when using all features or only filter methods. From Table 6, it’s clear that the proposed hybrid methods IG-DE with SVM (98%, 100%, and 98%), IGR-DE with DT (98%, 100%, and 98%), and Gini Index GIND-DE with DT (98%, 100%, and 98%) accomplished about the same performance in terms of accuracy, precision, and F-measure. In terms of selected features, as shown in Table 6, the hybrid GIND-DE with DT and IGR-DE with DT showed excellent performance using only 296 features out of a total of 12,600 features. In conclusion, the hybrid Gini Index and IGR with DE optimization and the DT classifier outperformed other hybrid methods.

In addition, Table 6 reveals that the filter feature selection strategy reduced the number of relevant features in the LUNG datasets from 12,600 features to 630 features. Furthermore, the suggested hybrid IG-DE, IGR-DE, CHSQR-DE, CR-DE, GIND-DE, and RELIEF-DE techniques reduced the 630 features to just 300, 305, 318, 308 and 294 relevant features on average. The smallest number of the selected features are highlighted in grey colour.

As described in Table 1, the Lung dataset has five classes: normal tissue (17 samples), adenocarcinoma (139 samples), pulmonary carcinoid (20 samples), squamous carcinoma (21 samples) and small cell cancer (6 samples). That means Lung consists of 17 samples of normal tissue, while remaining classes represent types of lung cancer. Therefore, in addition to classification measures, since there are different numbers of samples for classes in lung dataset, the classification errors in terms of False Positive Rate (FPR) and False Negative Rate (FNR) were calculated for each class in Table 7.

The FPR and FNR were calculated for SVM, NB, KNN, DT, and RF after applying the proposed hybrid filter-DE feature section methods: IG-DE, IGR-DE, CHSQR-DE, CR-DE, GIND-DE and RELIEF-DE. The lower FPR and FNR indicate less misclassification and better performance. As can be observed from Table 7, in most cases, TPRs and TNRs of SVM, NB, KNN, DT, and RF after applying the proposed hybrid filter-DE feature section methods were low, especially with the four types of lung cancer (adenocarcinoma, pulmonary carcinoid, squamous carcinoma and small cell cancer). Particularly, NB achieves low FPR across most feature selection methods, especially with IG-DE and IGR-DE. However, it struggles with higher FNR in Adenocarcinoma using CR-DE, impacting sensitivity for this class. DT maintains low FPR, with IGR-DE showing excellent results, though FNR for Adenocarcinoma remains moderately high across methods. IGR-DE provides the best overall balance for DT performance. RF achieves low FPR with IG-DE but shows slightly higher FPR for Normal tissue with methods like CHSQR-DE. FNR is highly variable, with challenges in Adenocarcinoma and Pulmonary Carcinoid depending on the feature selection method. KNN has consistently low FPR, particularly with IG-DE, though CHSQR-DE raises FPR for Normal tissue. FNR is more variable, with Pulmonary Carcinoid being challenging in certain feature selection methods. SVM shows high FPR in Normal tissue using IG-DE but maintains low FPR with other methods like IGR-DE. FNR is generally low, though Pulmonary Carcinoid can be challenging for GIND-DE and RELIEF-DE. We can conclude that NB and DT with IG-DE or IGR-DE feature selection offer the best balance of low FPR and FNR across most cancer types, ensuring minimal misclassifications. So, NB and DT with IG-DE or IGR-DE can be considered as effective options for this lung cancer dataset.

### 4.5. Results Analysis of Breast Dataset

Figure 5 shows that adding IG improved the classification accuracy of KNN (56.67%), NB (48.45%), DT (53.73%), RF (63.92%), and SVM (52.58%) to 71.33%, 55.67%, 67.01%, 86.6%, and 74.23% respectively. Furthermore, using the IGR filter method increased the accuracy results to 64.95%, 54.67%, 61.86%, 87.63%, and 69.07%, respectively. Additionally, in the case of CHSQR, the observed enhancement in comparison to all feature performances reached 72.16%, 72.33%, 68.04%, 81.44%, and 73.20%, respectively. In the case of CR (correlation), enhancement reached 76.29%, 77.32%, 67.01%, 76.29%, and 74.23%, respectively. In the case of GIND (Gini Index), enhancement in comparison to all feature performance reached 73.2%, 56.64%, 70.1%, 70.10%, 82.47%, and 78.35%, respectively. In the case of GIND (Gini Index), enhancement in comparison to all feature performance reached 73.2%, 56.64%, 70.1%, 70.10%, 82.47%, and 78.35%, respectively.

The relief filter also showed enhancement in the accuracy results and reached 74.23%, 77.32%, 63.92%, 80.41%, and 76.29%. By applying the filter method, the highest accuracy achieved was 87.63% with IGR-RF, and the lowest accuracy achieved was 54.67% with IGR-NB. The proposed hybrid IG-DE method improved the accuracy of KNN, NB, DT, RF, and SVM even more, to 80%, 53%, 87%, 70%, and 73%, respectively. The hybrid IGR-DE method improved them even more, to 70%, 53%, 87%, 70%, and 67%, respectively. The CHSQR-DE method enhanced further the classification accuracies of KNN, NB, DT, RF, and SVM to 77%, 70%, 93%, 80%, and 73%, respectively. By applying the hybrid CR-DE method, KNN, NB, DT, RF, and SVM results were enhanced to 83%, 70%, 87%, 70%, and 70%, respectively. Using the hybrid GIND-DE method, the accuracy of KNN, NB, DT, RF, and SVM improved to 80%, 60%, 87%, 77%, and 73%, respectively. Using the hybrid RELIEF-DE method, the accuracy also improved to 80%, 77%, 90%, 77%, and 73%, respectively. The highest accuracy (93%) enhancement was observed with CHSQR-DE with the DT classifier, while the lowest enhancement (53%) was observed with IG-DE and IGR-DE using the NB classifier. In terms of accuracy, the CHSQR-DE with the DT classifier achieved an optimal performance of 93% using the top 5% of ranked features on this Breast dataset.

Table 8 shows the performance measures, and the number of genes identified by the proposed methodology on the Breast dataset using all features, filters, and the proposed hybrid filter-DE methods. Table 8 clearly reveals that the machine learning algorithms performed significantly better in terms of accuracy, precision, recall, and F-measure after applying the proposed hybrid filter-DE methods than when using all features or only filter methods. Further, Table 8 shows that the suggested hybrid CHSQR-DE with DT outperformed other hybrid approaches in terms of accuracy, precision, and F-measure 93%, 94%, 93% respectively.

Furthermore, the relief RELIEF-DE with the DT classifier obtained accuracy, precision, and F-measures of 90%, 90%, and 90%, respectively. In terms of selected features, as shown in Table 8. CHSQR-DE with DT showed the best performance using the least number of features (615) out of a total of 24,481 features. In conclusion, Chi-Square with DE optimization and a DT classifier outperformed all other filter and hybrid methods. In addition, Table 8 indicates that the filter feature selection strategy reduced the number of relevant features in the BREAST datasets from 24,481 features to 1224 features. Further, the application of suggested hybrid CHSQR-DE-DT, CR-DT-KNN, CHSQR-DE-RF, and RELIEF-DE-NB techniques decreased the 1224 features to an average of 615, 583, 619 and 596 significant features, respectively. The smallest number of selected features are highlighted in grey color.

### 4.6. Analysis of Computational Time Complexity

To study computational complexity, big O notation was used in this study to analyze the computational time complexity of the proposed method. The computational times for Information gain (IG), information gain ratio (IGR), correlation (CR), Gini index (GIND), Relief (RELIEF) and Chi-squared (CHSQR) are O(F×N×log⁡N), O(F×N×log⁡N), O(F×N), O(F×N×log⁡N), O(l×F×N), and O(F×N), respectively. Here, F represents the number of features, N denotes the total number of samples in the dataset, l represents number of samples chosen to evaluate the features in Relief (RELIEF).

The time complexity of one generation in DE involves the time complexity of the mutation process O(P×D), the crossover process O(P×D), the fitness evaluation and selection process O(P×T). The combined complexity per generation is $O\left(P\times D\right)+O\left(P\times D\right)+O\left(P\times T\right)=O(P\times (D+T))$. Thus, the total time complexity of DE for all generations is O(G×P×(D+T)), where G is the total number of generations, P is the population size, D is the dimensionality of the problem (number of features reduced by filter methods), and T is the time complexity of evaluating the fitness function.

The overall time complexity of proposed hybrid approach involves calculating the computational time of filter algorithms used to rank the features and then reduce the dimensionality of microarray datasets, and the computational time of the DE optimization algorithm used in the second phase to identify the best features. Therefore, the total computational time complexity of the proposed hybrid filter-DE feature selection approaches can be listed in the following Table 9.

### 4.7. Comparison of Proposed Hybrid Filter-DE with Previous Works

This section presents a comparison between the proposed hybrid filter-DE and previous research works that utilized hybrid feature selection techniques on microarray datasets.

This study aimed to reduce the dimensionality of four microarray datasets: Lung, CNS, Brain, and Breast. Hameed et al. [35] proposed using PCC-GA and PCC-BPSO approaches in microarray datasets to integrate Pearson’s Correlation Coefficient (PCC) with GA or BPSO. Almutiri et al. [29] enhanced cancer classification using fusion-based feature selection on four microarray datasets. In one recent work, the authors [17] used the hybrid feature selection approach to evaluate these four microarray datasets. They combined the Gini index (GI) and support vector machines (SVMs) with recursive feature elimination (RFE).

Recently, Ali et al. [23] employed hybrid GA-based feature selection on the same four microarray datasets used in this study, integrating genetic algorithm with filter methods. Table 10 shows the accuracy comparison of the proposed hybrid filter-DE against previous research works. The proposed hybrid filter-DE feature selection approaches were compared to related studies using the same datasets. Table 10 reveals that the proposed approaches outperformed most of the previous works on most of the datasets used in this study. For Brain dataset, it is evident from Table 10 that the suggested hybrid filter-DE feature selection approaches with NB performed well with 100% accuracy. Furthermore, most of the proposed hybrid filter-DE methods with DT achieved 100% classification accuracy. Similarly, the classification accuracy results accomplished by previous work [23] with RF were competitive with our proposed methods. For CNS dataset, the classification accuracy results of most of the proposed hybrid filter-DE methods were better than accuracy results achieved by other works. Particularly, Table 10 reveals that the proposed IG-DE, CHSQR-DE, CR-DE, GIND-DE, and RELIEF-DE with NB for CNS dataset performed well with 100% accuracy. For Lung dataset, the suggested IGR-DE and GIND-DE with DT performed well with 98% accuracy. The IG-GA [23] and PCC-BPSO [35] with NB also performed comparable performance about 98% classification accuracy. For Breast dataset, it is evident from Table 10 that the proposed CHSQR-DE with the DT and IGR-GA [23] with RF outperformed all other approaches with 93% accuracy.

To show the improvement achieved, the improvement percentage of accuracy achieved by the proposed hybrid filter-DE methods is calculated using the following Formula (28):(28)$${IP}_{Acc}=\frac{{{Acc}_{S\_F}-Acc}_{All\_F}}{{Acc}_{All\_F}}\times 100$$
where ${IP}_{Acc}$ is the improvement percentage of accuracy, ${Acc}_{S\_F}$ is the accuracy achieved with features selected by the proposed hybrid filter-DE methods, and ${Acc}_{All\_F}$ the accuracy achieved with all features.

Table 11 shows the comparison of the improvement percentage of accuracy achieved by the proposed hybrid filter-DE methods and the previous studies. For Brain dataset, the average improvement percentage of accuracy achieved by the proposed hybrid filter-DE methods was up to 42.47%, while the previous works achieved the average improvement percentage of accuracy up to 41.43%. For CNS, the average improvement percentage of accuracy achieved by the proposed hybrid filter-DE methods was up to 57.45% while the average improvement percentage of previous works was up to 53.66%. For LUNG, the average improvement percentage of accuracy achieved by the proposed hybrid filter-DE methods was up to 16.28%, while the average improvement percentage for previous works was up to 17.53%. For BREAST, the average improvement percentage of accuracy achieved by the proposed hybrid filter-DE methods was up to 43.57%, while the average improvement percentage for previous works is up to 61.70%. We conclude the proposed hybrid filter-DE methods accomplished better improvement percentage of accuracy than the previous works in Brain and CNS datasets. For Lung dataset, the improvement percentage of accuracy achieved by the proposed hybrid filter-DE methods was competitive to the improvement percentage of accuracy achieved by the previous work. For BREAST dataset, the improvement percentage of accuracy achieved by the proposed hybrid filter-DE methods was less than the improvement percentage of accuracy achieved by the previous works.

### 4.8. Statistical Significance Testing

We have performed the Wilcoxon signed-rank tests, assuming the null hypothesis (H0) and alternative hypothesis (H1). The null hypothesis states that there is no difference between the accuracies of the pairs, while the alternative hypothesis states that there is a significant difference between the accuracies of the pairs. The Wilcoxon signed-rank test yields two values: the test statistic and the *p*-value. The *p*-value represents the likelihood of obtaining a result that is at least as extreme as the observed outcome under the assumption that the null hypothesis holds. The statistic uses ranks of differences between paired data. More minor statistics indicate more substantial evidence of paired sample differences. If *p* ≤ 0.05, there is significant evidence to reject the null hypothesis, and it shows significant differences between samples. If *p* > 0.05, we do not reject the null hypothesis, indicating no significant difference in performance and the two methods perform competitively.

#### 4.8.1. Statistical Significance Testing of Proposed Methods Compared with Each Other

From Table 12, we observed that the *p*-values of IG-DE vs. IGR-DE (*p* = 0.07), IG-DE vs. CR-DE (*p* = 0.08), CR-DE vs. GIND-DE (*p* = 0.16) and CR-DE vs. RELIEF-DE (*p* = 0.12) found very close to the significant level (0.05). This marginal difference validates that IG-DE and CR-DE performed in almost significant ways in competition.

The comparisons among IG-DE vs. CHSQR-DE, IG-DE vs. GIND-DE, IG-DE vs. RELIEF-DE, IGR-DE vs. CR-DE, CHSQR-DE vs. CR-DE, CHSQR-DE vs. GIND-DE, CHSQR-DE vs. RELIEF-DE, GIND-DE vs. RELIEF-DE were having high *p*-values (mostly > 0.05). For example, the comparison between IG-DE and CHSQR-DE yielded a *p*-value of 0.90, while the comparison between GIND-DE and RELIEF-DE produced a *p*-value of 0.73. No statistically significant differences exist in performances for IG-DE vs. GIND-DE, IG-DE vs. RELIEF-DE, IGR-DE vs. CR-DE, CHSQR-DE vs. CR-DE, CHSQR-DE vs. GIND-DE and CHSQR-DE vs. RELIEF-DE pairs. The high *p*-values validate that the null hypothesis (no difference in the method performance) cannot be rejected. This concludes that the performances of IG-DE vs. CHSQR-DE, IG-DE vs. GIND-DE, IG-DE vs. RELIEF-DE, IGR-DE vs. CR-DE, CHSQR-DE vs. CR-DE, CHSQR-DE vs. GIND-DE, CHSQR-DE vs. RELIEF-DE, GIND-DE vs. RELIEF-DE were not distinguishably different from each other.

The results revealed a promising future for feature selection methodologies, with IGR-DE emerging as a frontrunner, outperforming GIND-DE, RELIEF-DE, and CHSQR-DE. Other methods, such as IG-DE, CHSQR-DE, CR-DE, and GIND-DE, RELIEF-DE, demonstrated similar performance, suggesting their potential interchangeability without significant differences in accuracy.

#### 4.8.2. Statistical Significance Testing of Proposed Methods Against Other Existing Works

As shown in Table 13, the Wilcoxon signed-rank test showed that IGR-DE outperformed PCC-GA [35] (*p* = 0.01) and GI-SVM-RFE [26] (*p* = 0.006). IGR-DE outperforms GI-SVM-RFE [26] due to its lower *p*-values, specifically 0.006. The lower *p*-value (0.01) indicates that IGR-DE has better performance than PCC-GA [35]. The lower *p*-values (0.006 and 0.01) support the rejection of the null hypothesis, which states that there is no significant difference between the compared methods. These findings support our proposed IGR-DE approach. The performance of IGR-DE differs significantly from GI-SVM-RFE [26] and PCC-GA [35]. IGR-DE outperformed GI-SVM-RFE [26] and PCC-GA [35] due to this considerable difference.

Despite the slight difference in significance level, IGR-DE demonstrated a performance that was on par with other methods such as IG-GA [23] (*p* = 0.13), Fusion [29] (*p* = 0.10), and PCC-BPSO [35] (*p* = 0.11). The *p*-value is larger than 0.05 indicates that the proposed methods are competitive with previous research works. IGR-DE was found to be competitive with IGR-GA [23] (*p* = 1.0) and CS-GA [23] (*p* = 0.40), although not significantly different.

The Wilcoxon signed-rank test revealed that IG-DE performed comparably to PCC-GA [35] (0.09), PCC-BPSO [35] (0.11), and GI-SVM-RFE [26] (0.11) due to a minor variation in significant *p*-value. Even though IG-DE showed no significant difference from IG-GA [23] (*p* = 0.75), Fusion [29] (*p* = 0.68), IGR-GA [23] (*p* = 0.59), and CS-GA [23] (*p* = 0.98), the *p*-value is greater than 0.05 that implies the proposed approaches perform competitively to the existing works.

The Wilcoxon signed-rank test showed that CHSQR-DE outperformed PCC-GA [35] (*p* = 0.02) and PCC-BPSO [35] (*p* = 0.03). The lower *p*-value of CHSQR-DE at 0.02 indicates that the performance of CHSQR-DE is significantly superior to that of PCC-GA [35]. Further, the lower *p*-value of CHSQR-DE at 0.03 shows that CHSQR-DE outperforms PCC-BPSO [35]. Lower *p*-values (0.02 and 0.03) support null hypothesis rejection. These results validate our proposed CHSQR-DE methodology. The performance of CHSQR-DE differs significantly from PCC-GA [35] and PCC-BPSO [35]. CHSQR-DE outperformed PCC-GA [35] and PCC-BPSO [35] due to this considerable difference. The proposed method CHSQR-DE showed no significant difference compared to other existing methods such as IG-GA [23] (*p* = 0.54), Fusion [29] (*p* = 0.56), IGR-GA [23] (*p* = 0.45), CS-GA [23] (*p* = 0.34), and GI-SVM-RFE [26] (0.21), but it was still competitive because the *p*-value was higher than 0.05.

The analysis using the Wilcoxon signed-rank test revealed that CR-DE demonstrated a statistically significant improvement over PCC-GA [35] (*p* = 0.009) and PCC-BPSO [35] (*p* = 0.02). The lower *p*-value of CR-DE, at 0.009, means the proposed CR-DE outperforms the PCC-GA [35], while for CR-DE at 0.02, it indicates superior performance than PCC-BPSO [35]. Lower *p*-values (0.009 and 0.02) support null hypothesis rejection. These results support our proposed CR-DE approach. The performance of CR-DE differs significantly from PCC-GA [35] and PCC-BPSO [35]. The proposed CR-DE outperformed PCC-GA [35] and PCC-BPSO [35] due to this considerable difference. The proposed method CR-DE showed no significant difference compared to other previous research works like IG-GA [23] (*p* = 0.23), Fusion [29] (*p* = 0.84), IGR-GA [23] (*p* = 0.27), CS-GA [23] (*p* = 0.29), and GI-SVM-RFE [26] (*p* = 0.46), but it was still competitive because the *p*-value was higher than 0.05.

The results of the Wilcoxon signed-rank test showed that GIND-DE outperformed GI-SVM-RFE [26] with a *p*-value of 0.05. The *p*-value is a measure of the strength of the evidence against the null hypothesis. In this case, a *p*-value of 0.05 suggests that there is a 5% chance that the observed difference in performance between GIND-DE and GI-SVM-RFE [26] is due to random variation. GIND-DE outperformed GI-SVM-RFE [26] due to its lower *p*-values, precisely 0.05. Null hypothesis rejection is strengthened by the lower *p*-value of 0.05. This confirms the novel characteristics of our GIND-DE technique. The performance of GIND-DE differs significantly from GI-SVM-RFE [26]. The proposed GIND-DE approach showed it is about to perform with a significant difference from PCC-GA [35] (*p* = 0.07) and PCC-BPSO [35] (*p* = 0.09). The significant *p*-value is ≤0.05, and the proposed GIND-DE achieved a *p*-value of 0.07 and 0.09. This marginal difference demonstrates that GIND-DE performed in an almost significant way and very competitively compared to PCC-GA [35] and PCC-BPSO [35]. The proposed method GIND-DE showed no significant difference compared to the other existing methods like IG-GA [23] (*p* = 0.84), Fusion [29] (*p* = 0.56), IGR-GA [23] (*p* = 0.59), and CS-GA [23] (*p* = 0.75), but it was still competitive to them.

The Wilcoxon signed-rank test showed that RELIEF-DE outperformed PCC-GA [35] and PCC-BPSO [35] (*p* = 0.05). RELIEF-DE outperformed PCC-GA [35] and PCC-BPSO [35] because of its lower *p*-values, particularly 0.05. Null hypothesis rejection is strengthened by the lower *p*-value of 0.05. This result validates our novel RELIEF-DE methodology. The performance of RELIEF-DE differed significantly from PCC-GA [35] and PCC-BPSO [35]. RELIEF-DE outperformed PCC-GA [35] and PCC-BPSO [35] due to this considerable difference. The proposed approach RELIEF-DE achievement was found to be very close to differ significantly from GI-SVM-RFE [26] (*p* = 0.06). The observed *p*-value (0.06) is marginally different from 0.05. Consequently, RELIEF-DE was almost about to perform differently with GI-SVM-RFE [26]. The proposed method RELIEF-DE showed no significant difference compared to other previous research works such as IG-GA [23] (*p* = 0.64), Fusion [29] (*p* = 0.56), IGR-GA [23] (*p* = 0.49), and CS-GA [23] (*p* = 0.59). However, it remains competitive, as the *p*-value of higher than 0.05 indicates that the performance of the proposed method is on par with the existing works.

### 4.9. Discussion

Due to the dimensionality curse on four cancerous microarray datasets, the common machine learning algorithms trained with all features failed to produce outstanding classification results, as shown in Figure 2, Figure 3, Figure 4 and Figure 5 and Table 4, Table 5, Table 6, Table 7 and Table 8. To reduce the high dimension of microarray datasets, filter methods contributed to removing redundant and irrelevant features in order to improve classification results. For the majority of the microarray datasets utilized in this work, Figure 2, Figure 3, Figure 4 and Figure 5 and Table 4, Table 5, Table 6, Table 7 and Table 8 demonstrate that the IG, IGR, and CHSQR filter methods improved the performance of SVM, NB, kNN, DT, and RF. However, the filter approaches often assess features without consulting a classifier and disregard the relationship between the features sets and the class label. Therefore, SVM, NB, KNN, DT, and RF with filter techniques only produced modest classification improvements on cancerous microarray datasets.

Figure 2, Figure 3, Figure 4 and Figure 5 and Table 4, Table 5, Table 6, Table 7 and Table 8 demonstrate that when the proposed hybrid filter-DE feature selection techniques were applied, SVM, NB, kNN, DT, and RF performed noticeably better cancer classification results compared to their performances without applying feature selection or applying only filter methods. Figure 6 illustrates that the classification accuracy results achieved by the proposed hybrid filter-DE over filter methods increased to 100%, 100%, 93% and 98% on four microarray datasets: Brain, CNS, Breast and Lung, respectively. This was expected since the proposed hybrid filter-DE feature selection method was able to find set of optimal features by considering the correlation between the sets of features and the class labels. In addition to enhancing the classification measures, Figure 7 and Table 4, Table 5, Table 6, Table 7 and Table 8 show that applying the suggested DE-based feature selection contributed to removing around 50% of the irrelevant features reduced using filter methods for four cancerous microarray datasets. That means that a smaller number of significant features selected by DE can be utilized for cancer classification on microarray datasets.

Compared to other existing hybrid feature selection works [23,24,26] the results in Table 10 demonstrate that, for the majority of the microarray datasets, the proposed hybrid filter and DE-based feature selection outperformed or was competitive to other previous works that suggested combining filter methods with genetic algorithm, particle swarm optimization, or other fusion methods. It can be observed from Table 10 that NB after applying the filter method and DE-based feature selection exceeded the competitor methods and accomplished exceptional performance on Brain and CNS datasets. For Lung and Breast datasets, DT with applying the proposed hybrid GIND-DE and CHSQR-DE outperformed other existing hybrid feature selection works while performances of other machine learning classifiers were competitive to their performances with other existing hybrid feature selection works, as shown in Table 10.

## 5. Conclusions and Future Work

This study suggests hybridization of filter and differential evolutionary (DE) algorithm-based feature selection methods to deal with challenges associated to high-dimensional microarray datasets. The suggested hybrid filter-DE feature selection approach initially identifies the top-ranked five percent (5%) significant features by means of IG, IGR, CHSQR, GIND, CR, and RELIEF in order to get rid of any irrelevant and redundant features from high-dimensional microarray datasets. The filter feature selection approaches alone do not perform well in cancer classification because they evaluate the features independently of the machine learning algorithm. So, in the next phase, the suggested approach further performed an optimization on the reduced dataset by means of DE to enhance cancer classification. The experimental results showed that the proposed approaches using differential evolutionary (DE) algorithms effectively eliminated approximately 50% of irrelevant features from the initially refined datasets using filter methods. This process ensured that only essential features remained for optimizing cancer classification performance. In addition, the hybrid filter-DE feature selection approaches demonstrated superior performance compared to stand-alone classifiers and filter algorithm-only classifiers. Furthermore, the suggested hybrid DE method consistently did better than other hybrid feature selection methods on most of the high-dimensional microarray datasets that were used in this study. Moving forward, selecting the optimal parameters using optimization methods may enhance the results. Furthermore, using resampling or class-weight learning techniques to tackle data imbalances in the microarray datasets could enhance the suggested hybrid filter-DE feature selection approach. In addition, future research endeavors will focus on enhancing cancer classification for high-dimensional microarray datasets by applying several filter feature selection approaches in combination with various evolutionary and swarm algorithms.

## Figures and Tables

**Figure 1 cancers-16-03913-f001:**
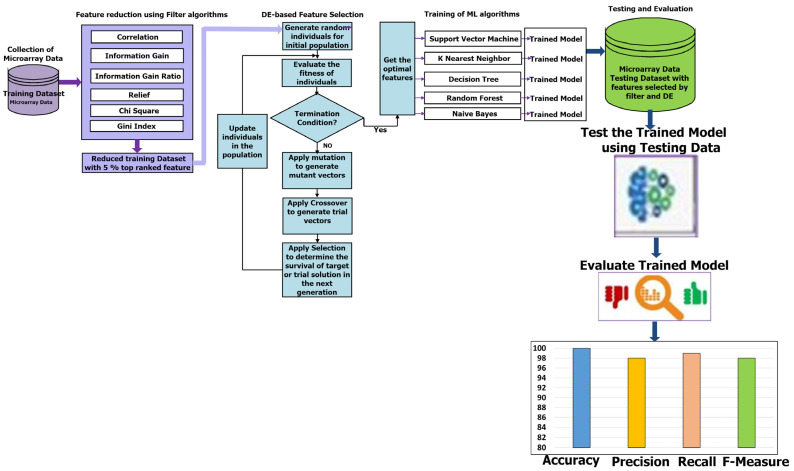
The methodology of improving cancer classification in Microarray data using the proposed hybrid filter and differential evolution-based feature selection.

**Figure 2 cancers-16-03913-f002:**
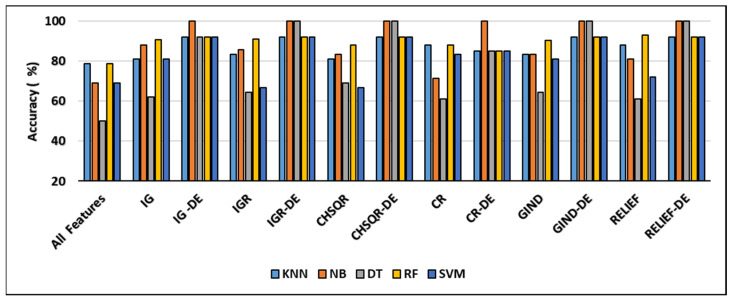
Comparison of machine learning accuracy on the Brain dataset using all features, filter features, and hybrid filter-DE methods.

**Figure 3 cancers-16-03913-f003:**
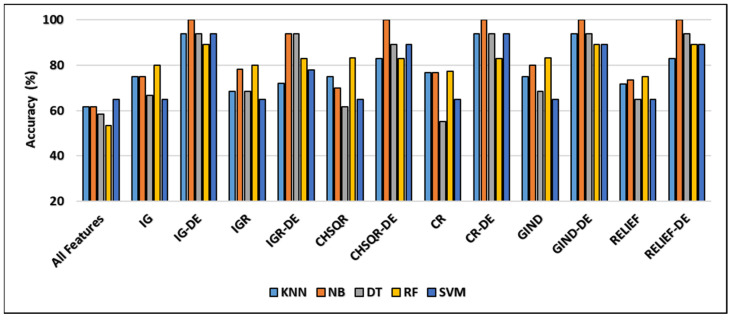
Comparison of machine learning accuracy on the CNS dataset using all features, filter features, and hybrid filter-DE methods.

**Figure 4 cancers-16-03913-f004:**
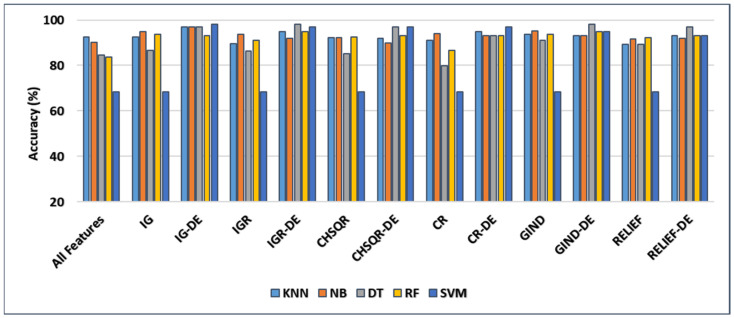
Comparison of machine learning accuracy on the Lung dataset using all features, filter features, and hybrid filter-DE features.

**Figure 5 cancers-16-03913-f005:**
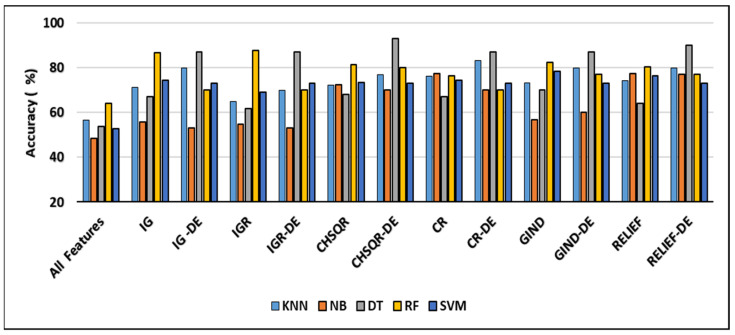
Comparison of accuracy results on the Breast dataset using all features, filter approaches, and the suggested hybrid filter-DE method.

**Figure 6 cancers-16-03913-f006:**
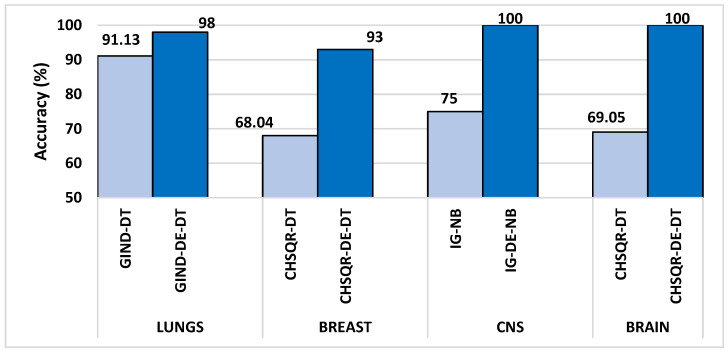
Comparison of the best accuracy results achieved by filter methods and the proposed hybrid filter-DE feature selection method.

**Figure 7 cancers-16-03913-f007:**
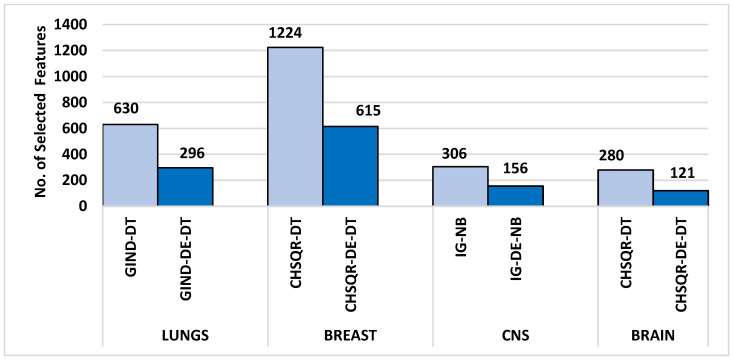
Comparison of the best results of features selected by filter methods and the proposed hybrid filter-DE feature selection method.

**Table 1 cancers-16-03913-t001:** Characteristics of the four cancerous microarray datasets used for assessing the proposed hybrid filter-DE feature selection method.

Dataset	No. of Samples	No. of Features	No. of Classes	Details of Classes
Breast [38]	97	24,481	2	Tumour (46 samples) and Normal (51 samples)
Lung [39]	203	12,600	5	The dataset consists of 17 samples of normal tissue, 139 samples of adenocarcinoma, 20 samples of pulmonary carcinoid, and 21 samples of squamous carcinoma and there are 6 samples of small cell cancer.
Brain [36,37]	42	5597	5	Medulloblastomas (10 samples), Non-embryonal brain tumours (10 samples), Normal human cerebella (4 samples), PNET (8 samples), and RERCNS (10 samples)
CNS [39]	60	7129	2	Tumour (39) and Normal (21)

**Table 2 cancers-16-03913-t002:** Some scenarios used to get the best crossover rate (CR) and population size (P).

		DE Parameters	Accuracy
**BREAST**	DE-CR-KNN	**P = 50, CR = 0.9**	** 83 **
		P = 100, CR = 0.5	73
		P = 150, CR = 0.6	73
	DE_CHSQR_DT	**P = 50, CR = 0.9**	**93**
		P = 100, CR = 0.5	67
		P = 150, CR = 0.6	63
		P = 200, CR = 0.7	63
**BRAIN**	DE-IG-NB	**P = 100, CR = 0.9**	**100**
		P = 150, CR = 0.6	85
		P = 100, CR = 0.5	85
		P = 200, CR = 0.7	92
	DE-IGR-NB	**P = 50, CR = 0.9**	**100**
		P = 200, CR = 0.7	77
		P = 150, CR = 0.6	77
		P = 100, CR = 0.5	69
	DE-IG-DT	**P = 50, CR = 0.9**	**92**
		P = 100, CR = 0.5	54
		P = 150, CR = 0.6	54
		P = 200, CR = 0.7	62
**CNS**	DE-IG-SVM	**P = −50, CR = 0.9**	**94**
		P = 100, CR = 0.5	93
		P = 150, CR = 0.6	94
		P = 200, CR = 0.7	94
	DE-GIND-NB	**P = 50, CR = 0.9**	**100**
		P = 100, CR = 0.5	89
		P = 150, CR = 0.6	83
		P = 200, CR = 0.7	94
	DE-RELIEF-DT	**P = 50, CR** **=** **0.9**	**94**
		P = 100, CR = 0.6	61
		P = 150, CR = 0.7	78
		P = 200, CR = 0.5	67
**LUNG**	DE-IGR-RF	**P = 50, CR = 0.9**	**95**
		P = 100, CR = 0.5.	90
		P = 150, CR = 0.7	92
		P = 200, CR = 0.7	93
	DE-GIND-DT	**P = 50, CR = 0.9**	**98**
		P = 100, CR = 0.5	77
		P = 150, CR = 0.6	82
		P = 200, CR = 0.7	82

**Table 3 cancers-16-03913-t003:** Parameter details of DE used in the hybrid filter-DE feature selection approach across the experiment on all datasets.

DE Parameters	Value
Crossover rate (CR)	0.9
Population size(P)	50
No of generations	50 to 600
Step size for number of generations	50

**Table 4 cancers-16-03913-t004:** Performance comparison of classifiers utilizing all features, filter methods, and the suggested hybrid filter-DE approaches on the Brain dataset. Background color: show the lowest number of features.

Brain		ALL- FT	IG	IG-DE	IGR	IGR-DE	CHSQR	CHSQR-DE	CR	CR-DE	GIND	GIND-DE	RELIEF	RELIEF-DE
SVM	Accuracy	69	80.95	92	66.67	92	66.67	92	83.33	85	80.95	92	71.9	92
	Recall	58	71	92	56	92	56	93	73	85	71	93	52	92
	Precision	48.28	66.18	94	47.47	94	45.18	95	67.09	88	66.18	95	47.69	94
	F-measure	52.7	68.51	92	51.38	92.99	50.01	93	69.92	86.47	68.51	93	49.75	92.99
	No. of Features	5597	280	129	280	142	280	137	280	131	280	140	280	149
NB	Accuracy	69.05	88.1	100	85.71	100	83.33	100	71.43	100	83.33	100	80.95	100
	Recall	60.05	80.5	100	78	100	76.05	100	63	100	73.05	100	72	100
	Precision	58.69	90.91	100	88.72	100	88.77	100	61.43	100	68.95	100	70	100
	F-measure	59.58	85.39	100	83.02	100	82.18	100	62.21	100	71.15	100	70.99	100
	No. of Features	5597	280	153	280	153	280	137	280	134	280	146	280	149
KNN	Accuracy	78.57	80.95	92	83.33	92	80.95	92	88.1	85	83.33	92	88.1	92
	Recall	75.05	80.5	93	83	93	80.5	93	86	87	83	93	88	93
	Precision	86.36	84.33	95	83.33	95	87.05	95	90.71	92	85.85	95	91.21	95
	F-measure	80.57	82.37	93	85.58	93	83.65	93	88.29	85	84.4	93	89.58	93
	No. of Features	5597	280	131	280	142	280	138	280	127	280	146	280	149
DT	Accuracy	50	61.9	92	64.29	100	69.05	100	61.09	85	64.29	100	61.9	100
	Recall	48.5	65	93	61.05	100	69	100	60	87	67.5	100	65	100
	Precision	53.06	61.73	95	63.83	100	69.89	100	60.16	85	65.73	100	64.95	100
	F-measure	50.68	63.32	93	62.64	100	69.44	100	60.08	82	66.6	100	64.97	100
	No. of Features	5597	280	149	280	145	280	121	280	134	280	139	280	122
RF	Accuracy	78.57	90.48	92	92.86	92	88.1	92	88.1	85	90.46	92	92.86	92
	Recall	76.5	88	92	93	92	88	93	83	87	88	93	93	92
	Precision	80.21	92.73	94	94.36	94	91.33	95	89.52	90	91.99	95	94.36	94
	F-measure	78.31	90.3	92	93.68	92	89.63	93	86.14	84	89.95	93	93.68	92
	No. of Features	5597	280	149	280	149	280	138	280	134	280	150	280	122

**Table 5 cancers-16-03913-t005:** Performance comparison of classifiers utilizing all features, filters, and the suggested hybrid filter-DE approaches on the CNS dataset. Background color: show the lowest number of features.

CNS		ALL- FT	IG	IG-DE	IGR	IGR-DE	CHSQR	CHSQR-DE	CR	CR-DE	GIND	GIND-DE	RELIEF	RELIEF-DE
SVM	Accuracy	65	65	94	65	78	65	89	65	94	65	89	65	89
	Recall	100	100	94	100	78	100	83	100	94	100	89	100	89
	Precision	65	65	95	65	77	65	93	65	95	65	90	65	90
	F-measure	78.79	78.89	94	100	77.5	78.79	86	78.79	94	78.79	88	78.79	89.5
	No. of Features	6129	306	156	306	170	306	176	306	176	306	175	306	178
NB	Accuracy	61.67	75	100	78.33	94	70	100	76.67	100	80	100	73.33	100
	Recall	66.67	76.92	100	84.62	92	71.79	100	79.49	100	82.05	100	76.92	100
	Precision	72.22	83.33	100	82.5	96	80	100	83.78	100	86.49	100	81.08	100
	F-measure	69.33	80	100	83.54	93	75.68	100	81.58	100	84.21	100	78.95	100
	No. of Features	6129	306	156	306	165	306	168	306	180	306	181	306	178
KNN	Accuracy	61.67	75	94	68.33	72	75	83	76.67	94	75	94	71.67	83
	Recall	79.49	82.05	92	84.62	72	87.18	83	79.49	92	79.49	94	87.18	83
	Precision	67.39	80	96	71.74	71	77.27	83	83.78	96	81.58	95	73.91	83
	F-measure	72.94	81.01	93	77.65	69	81.93	83	81.58	93	80.52	94	80	83
	No. of Features	6129	306	156	306	176	306	168	306	176	306	175	306	178
DT	Accuracy	58.33	66.67	94	68.33	94	61.67	89	55	94	68.33	94	65	94
	Recall	66.67	76.92	96	79.49	96	67.39	92	61.54	96	79.49	96	69.23	96
	Precision	68.42	73.17	93	73.81	93	79.49	88	66.67	93	73.81	93	75	93
	F-measure	67.53	75	94	76.54	94	72.94	88	64	94	76.54	94	69.23	94
	No. of Features	6129	306	179	306	174	306	181	306	177	306	181	306	185
RF	Accuracy	53.33	80	89	80	83	83.33	83	77.33	83	83.33	89	75	89
	Recall	71.79	89.74	88	97.44	75	82.22	79	87.18	79	89.74	83	87.18	83
	Precision	62.22	81.4	88	77.55	90	94.87	82	75.56	82	85.37	93	77.27	93
	F-measure	66.67	85.37	88	86.36	78	88.1	80	80.95	80	87.5	86	87.18	86
	No. of Features	6129	306	171	306	174	306	193	306	193	306	158	306	175

**Table 6 cancers-16-03913-t006:** Performance comparison of classifiers utilizing all features, filters, and the suggested hybrid filter-DE approaches on the Lung dataset. Background color: show the lowest number of features.

Lung		ALL- FT	IG	IG-DE	IGR	IGR-DE	CHSQR	CHSQR-DE	CR	CR-DE	GIND	GIND-DE	RELIEF	RELIEF-DE
SVM	Accuracy	68.47	68.47	98	68.47	97	68.47	97	68.47	97	68.47	95	68.47	93
	Recall	20	20	97	20	93	20	93	20	92	20	77	20	73
	Precision	13.69	13.69	100	13.69	99	13.69	99	13.69	99	13.69	79	13.69	78
	F-measure	16.25	16.25	98	16.25	95	16.25	95	16.25	95	16.25	77	16.25	75
	No. of Features	12,600	630	301	630	304	630	321	630	309	630	290	630	295
NB	Accuracy	90.15	95	97	93.6	92	92.12	90	94.09	93	95.07	93	91.63	92
	Recall	79.07	88.5	97	93.64	92	93.21	90	90.37	93	88.5	93	88.53	92
	Precision	88.21	94.36	97	88.9	93	84.92	92	89.07	93	90.49	94	83.45	93
	F-measure	83.39	91.34	97	91.21	92	88.87	91	89.72	93	89.48	94	85.91	92
	No. of Features	12,600	630	301	630	304	630	321	630	309	630	290	630	295
KNN	Accuracy	92.61	92.61	97	89.66	95	92.12	92	91.13	95	93.6	93	89.16	93
	Recall	80.73	87.91	96	73.98	92	80.36	82	86.45	94	84.79	86	68.85	73
	Precision	95.22	89.01	96	93.1	95	95.09	91	86.96	93	93.38	93	93.38	78
	F-measure	87.38	88.5	96	82.45	93	87.11	85	86.7	95	88.88	87	79.26	75
	No. of Features	12,600	630	301	630	304	630	321	630	309	630	290	630	295
DT	Accuracy	84.43	86.7	97	86.21	98	85.22	97	79.8	93	91.13	98	89.16	97
	Recall	69.07	73.69	93	79.2	96	72.4	93	69.07	95	84.69	96	86.45	93
	Precision	84.15	80.63	99	80.13	100	85.92	99	75.92	90	83.49	100	86.43	99
	F-measure	75.87	77	95	79.66	98	78.58	95	72.33	92	84.09	98	86.44	95
	No. of Features	12,600	630	296	630	296	630	317	630	311	630	296	630	304
RF	Accuracy	83.74	93.6	93	91.13	95	92.61	93	86.7	93	93.6	95	92.12	93
	Recall	59.01	97.15	88	78.46	92	83.47	82	68.77	88	82.99	92	75.6	88
	Precision	93.54	85.6	94	94.45	94	96.82	94	96.75	94	97.07	95	95.96	94
	F-measure	72.43	91.01	91	85.72	93	89.65	86	80.4	91	89.48	93	84.57	91
	No. of Features	12,600	630	303	630	320	630	313	630	303	630	308	630	289

**Table 7 cancers-16-03913-t007:** The comparison of FPR and FNR for the suggested hybrid filter-DE approaches on the Lung dataset.

			Classes in Lung Dataset					
			Normal Tissue	Adeno- Carcinoma	Pulmonary Carcinoid	Squamous Carcinoma	Small Cell Cancer	Average
SVM	IG-DE	FPR	0.53	0.00	0.00	0.00	0	0.11
		FNR	0.00	0.00	0.00	0.17	0	0.03
	IGR-DE	FPR	0.11	0.00	0.00	0.00	0	0.02
		FNR	0.00	0.40	0.00	0.00	0	0.08
	CHSQR-DE	FPR	0.11	0.00	0.00	0.00	0	0.02
		FNR	0.00	0.20	0.00	0.17	0	0.07
	CR-DE	FPR	0.11	0.00	0.00	0.00	0	0.02
		FNR	0.00	0.40	0.00	0.00	0	0.08
	GIND-DE	FPR	0.16	0.00	0.00	0.00	0	0.03
		FNR	0.00	0.00	1.00	0.17	0	0.23
	RELIEF-DE	FPR	0.21	0.00	0.00	0.00	0	0.04
		FNR	0.00	0.20	1.00	0.17	0	0.27
NB	IG-DE	FPR	0.05	0.02	0.00	0.00	0	0.01
		FNR	0.02	0.20	0.00	0.00	0	0.04
	IGR-DE	FPR	0.05	0.02	0.00	0.05	0	0.02
		FNR	0.10	0.20	0.00	0.00	0	0.06
	CHSQR-DE	FPR	0.05	0.02	0.02	0.05	0	0.03
		FNR	0.12	0.20	0.00	0.00	0	0.06
	CR-DE	FPR	0.16	0.02	0.00	0.00	0	0.04
		FNR	0.02	0.40	0.00	0.17	0	0.12
	GIND-DE	FPR	0.05	0.02	0.00	0.04	0	0.02
		FNR	0.07	0.20	0.00	0.00	0	0.05
	RELIEF-DE	FPR	0.05	0.02	0.00	0.05	0	0.02
		FNR	0.10	0.20	0.00	0.00	0	0.06
KNN	IG-DE	FPR	0.05	0.02	0.00	0.00	0	0.01
		FNR	0.02	0.00	0.00	0.17	0	0.04
	IGR-DE	FPR	0.11	0.02	0.00	0.00	0	0.02
		FNR	0.02	0.20	0.00	0.17	0	0.08
	CHSQR-DE	FPR	0.16	0.02	0.00	0.02	0	0.04
		FNR	0.05	0.20	0.50	0.17	0	0.18
	CR-DE	FPR	0.05	0.02	0.00	0.02	0	0.02
		FNR	0.48	0.20	0.00	0.00	0	0.14
	GIND-DE	FPR	0.11	0.04	0.00	0.00	0	0.03
		FNR	0.05	0.00	0.50	0.17	0	0.14
	RELIEF-DE	FPR	0.21	0.00	0.00	0.00	0	0.04
		FNR	0.00	0.20	1.00	0.17	0	0.27
DT	IG-DE	FPR	0.11	0.00	0.00	0.00	0	0.02
		FNR	0.00	0.20	0.00	0.17	0	0.07
	IGR-DE	FPR	0.05	0.00	0.00	0.00	0	0.01
		FNR	0.00	0.20	0.00	0.00	0	0.04
	CHSQR-DE	FPR	0.11	0.00	0.00	0.00	0	0.02
		FNR	0.00	0.20	0.00	0.17	0	0.07
	CR-DE	FPR	0.05	0.04	0.00	0.02	0	0.02
		FNR	0.07	0.20	0.00	0.00	0	0.05
	GIND-DE	FPR	0.05	0.00	0.00	0.00	0	0.01
		FNR	0.00	0.20	0.00	0.00	0	0.04
	RELIEF-DE	FPR	0.11	0.00	0.00	0.00	0	0.02
		FNR	0.00	0.20	0.00	0.17	0	0.07
RF	IG-DE	FPR	0.02	0.02	0.00	0.00	0	0.01
		FNR	0.02	0.40	0.00	0.17	0	0.12
	IGR-DE	FPR	0.11	0.02	0.00	0.00	0	0.02
		FNR	0.02	0.40	0.00	0.00	0	0.08
	CHSQR-DE	FPR	0.16	0.02	0.00	0.00	0	0.03
		FNR	0.02	0.40	0.50	0.00	0	0.18
	CR-DE	FPR	0.16	0.02	0.00	0.00	0	0.03
		FNR	0.02	0.40	0.00	0.17	0	0.12
	GIND-DE	FPR	0.11	0.02	0.00	0.00	0	0.02
		FNR	0.02	0.20	0.00	0.17	0	0.08
	RELIEF-DE	FPR	0.16	0.02	0.00	0.00	0	0.04
		FNR	0.02	0.40	0.00	0.17	0	0.12

**Table 8 cancers-16-03913-t008:** Classifier performance comparison on the Breast dataset using all features, filters, and the proposed hybrid filter-DE methods. Background color: show the lowest number of features.

Breast		ALL- FT	IG	IG-DE	IGR	IGR-DE	CHSQR	CHSQR-DE	CR	CR-DE	GIND	GIND-DE	RELIEF	RELIEF-DE
SVM	Accuracy	52.58	74.23	73	69.07	73	73.2	73	74.23	73	78.35	73	76.29	73
	Recall	100	80.39	73	76.47	73	70.59	73	70.59	73	80.39	73	72.55	73
	Precision	52.58	73.21	74	68.42	74	76.6	75	78.26	73	78.85	73	80.43	74
	F-measure	68.92	76.64	73	72.22	73.5	73.47	73	74.23	73	80.39	73	76.29	73
	No. of Features	24,481	1224	591	1224	606	1224	617	1224	610	1224	639	1224	581
NB	Accuracy	48.45	55.67	53	54.64	53	72.33	70	77.32	70	56.64	60	77.32	77
	Recall	76.47	96.08	50	90.2	50	74.51	70	78.43	70	96.08	57	78.43	77
	Precision	50.65	55.44	27	54.12	27	73.08	70	78.43	70	53.85	79	78.43	77
	F-measure	60.94	69.5	35	67.65	35	73.79	70	78.43	70	96.08	49	78.43	77
	No. of Features	24,481	1224	623	1224	572	1224	572	1224	610	1224	576	1224	596
KNN	Accuracy	55.67	71.13	80	64.95	70	72.16	77	76.29	83	73.2	80	74.23	80
	Recall	78.43	82.35	80	90.2	68	84.31	76	76.47	83	88.24	80	82.35	80
	Precision	55.56	68.85	80	61.33	82	69.35	77	78	83	69.23	80	72.41	80
	F-measure	65.04	75	80	73.02	65	76.11	76	77.23	83	77.59	80	77.06	80
	No. of Features	24,481	1224	622	1224	612	1224	584	1224	583	1224	581	1224	622
DT	Accuracy	57.73	67.01	87	61.86	87	68.04	93	67.01	87	70.1	87	63.92	90
	Recall	66.67	72.55	86	68.63	87	68.63	93	78.43	87	76.47	87	68.63	90
	Precision	58.62	62.27	88	62.5	87	70	94	65.57	87	69.64	87	64.81	90
	F-measure	62.39	69.81	86	65.42	87	69.31	93	71.43	87	72.9	87	66.67	90
	No. of Features	24,481	1224	599	1224	623	1224	615	1224	603	1224	605	1224	595
RF	Accuracy	63.92	86.6	70	87.63	70	81.44	80	76.29	70	82.47	77	80.41	77
	Recall	70.59	88.24	70	90.2	70	84.31	80	78.43	70	86.27	77	78.43	77
	Precision	64.29	86.54	70	86.89	72	81.13	80	76.92	70	81.48	77	83.33	77
	F-measure	67.29	87.38	69	88.46	69	82.69	80	77.67	70	86.27	76	80.81	77
	No. of Features	24,481	1224	595	1224	619	1224	619	1224	604	1224	620	1224	596

**Table 9 cancers-16-03913-t009:** The computational time complexity of the proposed methods.

The Proposed Hybrid Methods	Time Complexity
IG-DE IGR-DE GIND-DE	$O\left(F\times N\times \log N\right)+O(G\times P\times (D+T))$
CR-DE CHSQR-DE	$O\left(F\times N\right)+O(G\times P\times (D+T))$
RELIEF-DE	$O\left(l\times F\times N\right)+O(G\times P\times (D+T))$

**Table 10 cancers-16-03913-t010:** The accuracy comparison of the proposed hybrid filter-DE methods with the previous studies.

		Proposed Hybrid Filter-DE						IG-GA [23]	IGR-GA [23]	CS-GA [23]	GI-SVM-RFE [26]	Fusion [29]	PCC-GA [35]	PCC-BPSO [35]
		IG-DE	IGR-DE	CHSQR-DE	CR-DE	GIND-DE	RELIEF-DE							
BRAIN	KNN	92	92	92	85	92	92	92.86	97.62	97.62	87.5	95	95.24	97.62
	NB	100	100	100	100	100	100	92.86	95.24	95.24	88	N/A	90.48	92.86
	DT	92	100	100	85	100	100	85.71	88.1	85.71	71.5	N/A	N/A	N/A
	RF	92	92	92	85	92	92	100	100	100	90	88.67	95.24	85.71
	SVM	92	92	92	85	92	92	85.71	97.62	97.62	N/A	N/A	97.62	97.62
CNS	KNN	94	72	83	94	94	83	93.33	83.33	88.33	81.67	N/A	96.67	93.55
	NB	100	94	100	100	100	100	90	88.33	83.33	85	N/A	90	91.94
	DT	94	94	89	94	94	94	93.33	93.33	88.33	75	N/A	N/A	N/A
	RF	89	83	83	83	89	89	91.67	90	88.33	83.33	76.48	85	91.94
	SVM	94	78	89	94	89	89	86.67	65	83.33	N/A	75	98.33	91.94
LUNG	KNN	97	95	92	95	93	93	97.04	96.06	95.57	92.62	N/A	97.54	96.06
	NB	97	92	90	93	93	92	98.52	97.54	97.04	91.17	N/A	97.04	98.03
	DT	97	98	97	93	98	97	96.55	96.06	96.55	88.71	N/A	N/A	N/A
	RF	93	95	93	93	95	93	96.06	95.57	96.06	93.64	N/A	96.06	96.06
	SVM	98	97	97	97	95	93	94.09	94.58	95.07	N/A	N/A	97.54	97.04
BREAST	KNN	80	70	77	83	80	80	89.69	86.6	84.54	87.67	N/A	86.6	87.63
	NB	53	53	70	70	60	77	57.37	62.89	79.38	90.67	N/A	85.57	88.66
	DT	87	87	93	87	87	90	86.6	90.72	84.54	72.22	N/A	N/A	N/A
	RF	70	70	80	70	77	77	89.69	93.81	85.57	88.67	84.65	84.54	85.57
	SVM	73	73	73	73	73	73	84.54	82.47	82.47	N/A	75.11	88.66	90.72

**Table 11 cancers-16-03913-t011:** The comparison of the improvement percentage of accuracy achieved by the proposed hybrid filter-DE methods and the previous studies.

		Proposed Hybrid Filter-DE						IG-GA [23]	IGR-GA [23]	CS-GA [23]	GI-SVM-RFE [26]	FUSION [29]	PCC-GA [35]	PCC-BPSO [35]
		IG-DE	IGR-DE	CHSQR-DE	CR-DE	GIND-DE	RELIEF-DE							
BRAIN	KNN	17.09	17.09	17.09	8.18	17.09	17.09	18.19	24.25	24.25	11.37	20.91	21.22	24.25
	NB	44.82	44.82	44.82	44.82	44.82	44.82	34.48	37.93	37.93	27.44		31.04	34.48
	DT	84.00	100.00	100.00	70.00	100.00	100.00	71.42	76.20	71.42	43.00			
	RF	17.09	17.09	17.09	8.18	17.09	17.09	27.28	27.28	27.28	14.55	12.85	21.22	9.09
	SVM	33.33	33.33	33.33	23.19	33.33	33.33	24.22	41.48	41.48			41.48	41.48
	Average	39.27	42.47	42.47	30.88	42.47	42.47	35.12	41.43	40.47	24.09	16.88	28.74	27.32
CNS	KNN	52.42	16.75	34.59	52.42	52.42	34.59	51.34	35.12	35.12	32.43		56.75	51.69
	NB	62.15	52.42	62.15	62.15	62.15	62.15	45.94	43.23	35.12	37.83		45.94	49.08
	DT	61.15	61.15	52.58	61.15	61.15	61.15	60.00	60.00	42.86	28.58			
	RF	66.89	55.63	55.63	55.63	66.89	66.89	71.89	68.76	56.25	56.25	43.41	59.38	72.40
	SVM	44.62	20.00	36.92	44.62	36.92	36.92	33.34	0.00	28.20		15.38	51.28	41.45
	Average	57.45	41.19	48.38	55.20	55.91	52.34	52.50	41.42	39.51	38.77	29.40	53.34	53.66
LUNG	KNN	4.74	2.58	0.66	2.58	0.42	0.42	4.78	3.73	3.20	0.01		5.32	3.73
	NB	7.60	2.05	0.17	3.16	3.16	2.05	9.28	8.20	7.64	1.13		7.64	8.74
	DT	14.89	16.07	14.89	10.15	16.07	14.89	14.36	13.77	14.36	5.07			
	RF	11.06	13.45	11.06	11.06	13.45	11.06	14.71	14.13	14.71	11.82		14.71	14.71
	SVM	43.13	41.67	41.67	41.67	38.75	35.83	37.42	38.13	38.85			42.46	41.73
	Average	16.28	15.16	13.36	13.72	14.37	12.85	16.11	15.59	15.75	4.51		17.53	17.23
BREAST	KNN	43.70	25.74	38.32	49.09	43.70	43.70	61.11	55.56	51.86	57.48		55.56	57.41
	NB	9.39	9.39	44.48	44.48	23.84	58.93	18.41	29.80	63.84	87.14		76.62	82.99
	DT	50.70	50.70	61.09	50.70	50.70	55.90	50.01	57.15	46.44	25.10			
	RF	9.51	9.51	25.16	9.51	20.46	20.46	40.32	46.76	33.87	38.72	32.43	32.26	33.87
	SVM	38.84	38.84	38.84	38.84	38.84	38.84	60.78	56.85	56.85		42.85	68.62	72.54
	Average	30.43	26.84	41.58	38.52	35.51	43.57	46.13	49.22	50.57	52.11	37.64	58.26	61.70

**Table 12 cancers-16-03913-t012:** Comparison of the *p*-value of the proposed methods with each other.

Proposed Methods	*p*-Value
IG-DE vs. IGR-DE	0.07
IG-DE vs. CHSQR-DE	0.90
IG-DE vs. CR-DE	0.08
IG-DE vs. GIND-DE	0.59
IG-DE vs. RELIEF-DE	0.95
IGR-DE vs. CHSQR-DE	**0.05**
IGR-DE vs. CR-DE	0.58
IGR-DE vs. GIND-DE	**0.02**
IGR-DE vs. RELIEF-DE	**0.02**
CHSQR-DE vs. CR-DE	0.29
CHSQR-DE vs. GIND-DE	0.47
CHSQR-DE vs. RELIEF-DE	0.31
CR-DE vs. GIND-DE	0.16
CR-DE vs. RELIEF-DE	0.12
GIND-DE vs. RELIEF-DE	0.73

**Table 13 cancers-16-03913-t013:** Comparison of the *p*-value of the proposed methods against previous works.

Proposed Method	Previous Works	*p*-Value
IG-DE	IG-GA [23]	0.75
	IGR-GA [23]	0.59
	Fusion [29]	0.68
	CS-GA [23]	0.98
	GI-SVM-RFE [26]	0.11
	PCC-GA [35]	0.09
	PCC-BPSO [35]	0.11
IGR-DE	IG-GA [23]	0.13
	IGR-GA [23]	1.0
	Fusion [29]	0.10
	CS-GA [23]	0.40
	GI-SVM-RFE [26]	**0.006**
	PCC-GA [35]	**0.01**
	PCC-BPSO [35]	0.11
CHSQR-DE	IG-GA [23]	0.54
	IGR-GA [23]	0.45
	Fusion [29]	0.56
	CS-GA [23]	0.34
	GI-SVM-RFE [26]	0.21
	PCC-GA [35]	**0.02**
	PCC-BPSO [35]	**0.03**
CR-DE	IG-GA [23]	0.23
	IGR-GA [23]	0.27
	Fusion [29]	0.84
	CS-GA [23]	0.29
	GI-SVM-RFE [26]	0.46
	PCC-GA [35]	**0.009**
	PCC-BPSO [35]	**0.02**
GIND-DE	IG-GA [23]	0.84
	IGR-GA [23]	0.59
	Fusion [29]	0.56
	CS-GA [23]	0.75
	GI-SVM-RFE [26]	**0.05**
	PCC-GA [35]	0.07
	PCC-BPSO [35]	0.09
RELIEF-DE	IG-GA [23]	0.64
	IGR-GA [23]	0.49
	Fusion [29]	0.56
	CS-GA [23]	0.59
	GI-SVM-RFE [26]	0.06
	PCC-GA [35]	**0.05**
	PCC-BPSO [35]	**0.05**

## Data Availability

Available upon request.

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
