# Peer review of "Enhancing Cancerous Gene Selection and Classification for High-Dimensional Microarray Data Using a Novel Hybrid Filter and Differential Evolutionary Feature Selection"

_cancers, 2024, doi:10.3390/cancers16233913_

Round 1

Reviewer 1 Report

Comments and Suggestions for Authors

This work proposed a hybrid filter and differential evolutionary feature selection to improve the cancerous gene selection and classification for high-dimensional microarray data. The level of detail is generally good in major parts of the article; however, there are some key issues to be addressed. I recommend a major revision to this article. Please find my comments below:
Comment 1. State the percentage improvements by the proposed work compared to existing works.
Comment 2. More terms should be added to the “Keywords” to increase the chance of paper downloading rate.
Comment 3. Refer to the journal’s template to update the style of references.
Comment 4. Section 1 Introduction:
(a) Clarify which types of cancers were considered. In addition, discuss their severity and prevalence.
(b) Regarding the research contributions, the authors should state the percentage improvement compared to the existing works. In addition, some information related to ablation experiments should be added.
Comment 4. Section 2 Related works:
(a) The literature review misses the latest references in 2023-2024.
(b) The last paragraph should provide in-text citations to indicate which existing works were referred to.
Comment 5. Enhance the resolutions of all figures. Zoom in the Word file to confirm that no content is blurred.
Comment 6. Section 3 Materials and Methods:
(a) Figure 1 causes confusion:
- Why was feature reduction performed before feature extraction and selection?
- Regarding the training of machine learning algorithms, the building block implies that ensembled learning was used.  

- The flows contain some mistakes, training of ML algorithm -> Testing and Evaluation -> Trained Model -> Evaluate.
(b) Align subsections using the order of workflow in Figure 1.
(c) Justify the method in Subsection 3.2.
(d) Many deep learning-based approaches exist; why were traditional machine learning algorithms (SVM, NB, kNN, DT, and RF) used in Subsection 3.4?
(e) Algorithm 1:
- Check the alignment in each line.
- Add inputs and outputs.
Comment 7. Section 4 Experimental Environmental and Results Discussion:
(a) Table 2:
- More scenarios should be tested for the crossover rate and population size.
- What is the step size for the number of generations?
(b) How did the authors optimally design the model using kNN, NB, DT, RF, and SVM?
(c) Regarding the results, it can be seen that the results using SVM were obviously poorer, which is not common in medical diagnosis. It seems that there were some issues during the implementation or evaluation.
(d) Did you use k-fold cross-validation?
(e) Many existing works adopted deep learning algorithms and their results were not perfect; why did NB achieve perfect performance?

Author Response

Response to Reviewer 1 Comments

Dear reviewer

We would like to express our gratitude to you for your valuable time and efforts you have spent in the review. Please find the detailed responses below and the corresponding corrections highlighted in the re-submitted files. We have carefully considered your suggestions and made some changes. We have tried our best to improve and make changes in the manuscript.

Sincerely Yours

Corresponding Authors

Dr. Arshad Hashmi

Department of Information Systems

Faculty of Computing and Information Technology-Rabigh

King Abdulaziz University

Kingdom of Saudi Arabia

Assoc. Prof. Dr. Waleed Ali

Department of Information Technology

Faculty of Computing and Information Technology- Rabigh

King Abdulaziz University

Kingdom of Saudi Arabia

Comment 1: State the percentage improvements by the proposed work compared to existing works.

Thank

Response 1: Thank you very much. As recommended, we added one sentence in the abstract and introduction section that stated the percentage improvements by the proposed work compared to existing works.

" The average improvement percentages of accuracy achieved by the proposed methods were up to 42.47%, 57.45%, 16.28% and 43.57% compared to the previous works that are 41.43%, 53.66%, 17.53%, 61.70% on Brain, CNS, Lung and Breast datasets, respectively"

Comment 2: More terms should be added to the “Keywords” to increase the chance of paper downloading rate.

Response 2: Thank you very much. As recommended, several Keywords have been added to the article

Cancer classification, Brain cancer, Breast cancer, Lung cancer, Central Nervous System  cancer, Machine learning

Comment 3: Refer to the journal’s template to update the style of references.

Response 3: As recommended, the references were updated based on the journal style.

Comment 4:  Section 1 Introduction:
(a) Clarify which types of cancers were considered. In addition, discuss their severity and prevalence.

Response 4.a: Thank you very much. As recommended, we described and discussed the four types of cancers and their severity and prevalence in introduction in lines 54-71

(b) Regarding the research contributions, the authors should state the percentage improvement compared to the existing works. In addition, some information related to ablation experiments should be added.

Response 4.b: Thank you very much. As recommended, we calculated the percentage improvements by the proposed work and the existing works in subsection 4.7. Comparison of Proposed Hybrid Filter-DE with Previous Works. Then, we added one sentence in abstract and introduction section that stated the percentage improvements by the proposed work compared to existing works.

" The average improvement percentages of accuracy achieved by the proposed methods were up to 42.47%, 57.45%, 16.28% and 43.57% compared to the previous works that are 41.43%, 53.66%, 17.53%, 61.70% on Brain, CNS, Lung and Breast datasets, respectively"

Comment 5: Section 2 Related works:
(a) The literature review misses the latest references in 2023-2024.

Response 5.a: Thank you very much. As recommended, the literature review has been updated with latest references

(b) The last paragraph should provide in-text citations to indicate which existing works were referred to.

Response 5.b:

Thank you very much for valuable comment. As can be observed from the existing works discussed above, the filter methods have been utilized individually [18,24] or combined with the genetic algorithm [23-24] [26-27][33-35] or wrapper feature selection [29] [32] in order to improve cancer classification on microarray datasets.

Comment 6: Enhance the resolutions of all figures. Zoom in the Word file to confirm that no content is blurred.

Response 6: Thank you very much. As recommended, we enhanced the resolutions of figures

Comment 7: Section 3 Materials and Methods:
(a) Figure 1 causes confusion:
- Why was feature reduction performed before feature extraction and selection?(

Response: Actually, we did not mention feature extraction in Figure since all the datasets used are numeric and the features already extracted and converted into numeric. If you refer to Table 1. Characteristics of the four cancerous microarray datasets, you will see the four data set features have too many features (For example: 24,481 features for Breast dataset). It will be time consuming to use DE-based feature selection with all features. Therefore, common and fast filter feature selection method are used to reduce the high dimensional datasets. Consequently, the most optimal features are further selected by DE as second phase to produce outstanding cancer classification results.

- Regarding the training of machine learning algorithms, the building block implies that ensembled learning was used.  

Response: Thank you very much. As recommended, we enhanced Figure to show that we used machine learning algorithms individually.

- The flows contain some mistakes, training of ML algorithm -> Testing and Evaluation -> Trained Model -> Evaluate.

Response:

I think there is misunderstanding of the diagram follows and the phase name. In Figure 1, we write the phase name of top and the details down.  The methodology of the proposed hybrid filter and DE-based feature selection consists of five phases, as shown in Figure 1: microarray data collection, feature reduction using filter algorithms, feature selection using DE, training, and testing and evaluation of trained model. In the testing and evaluation phase, the testing datasets are input to the trained model (that already trained in the previous phase training of ML) and then we evaluate output of the model using evaluation measures.

 (b) Align subsections using the order of workflow in Figure 1.

We did it as recommended.

3.1 Description of Cancerous Microarray Data Used

 3.2 Feature Reduction Using Filter Algorithms

3.2.1. Correlation

3.2.2. Information Gain

3.2.3. Information Gain Ratio

3.2.4. Relief

3.2.5. Chi-Squared

3.2.6. Gini Index  

 3.3 Differential Evolution Based Feature Selection 

 3.4 Training of Machine Learning Techniques

 3.5 Testing and Evaluation of Machine Learning Techniques

(c) Justify the method in Subsection 3.2.

Response: As recommended, the justification of selection of the filter methods have been mentioned in line 293-297

"In this work, the features of microarray cancer datasets are ranked with various common filter feature selection methods.  The features of microarray cancer datasets are ranked by utilizing well-known, fast and effective filtering methods such as correlation, information gain, and information gain ratio, Relief, Chi-squared and Gini Index."

(d) Many deep learning-based approaches exist; why were traditional machine learning algorithms (SVM, NB, kNN, DT, and RF) used in Subsection 3.4?

Response: Although neural networks (BPNN) and deep learning algorithms are common machine learning algorithms, it is time consuming to use with cancerous microarray datasets that have too many features. They will need many input nodes and more nodes in other layers causing too much computational process.  Furthermore, it is more time-consuming and impractical to combine deep learning with DE that needs also more iterations or generations.  

(e) Algorithm 1:
- Check the alignment in each line.

Response: Done as recommended in Algorithm 1

- Add inputs and outputs.

Response: Done as recommended in Algorithm 1

Comment 7. Section 4 Experimental Environmental and Results Discussion:
(a) Table 2:
- More scenarios should be tested for the crossover rate and population size.

Response:

 Thank you very much. We have done many experiments for many scenarios and selected the best results.

- What is the step size for the number of generations?

Response: We wrote Step size for number of generations (50) in Table 2

 (b) How did the authors optimally design the model using kNN, NB, DT, RF, and SVM?

Response: The best settings and parameters used in all classifiers were selected by a trial-and-error basis in order to produce the best results. However, selecting the optimal parameters using optimization methods may enhance the results. We wrote one sentence in Section conclusion and Future work stated that issue.

(c) Regarding the results, it can be seen that the results using SVM were obviously poorer, which is not common in medical diagnosis. It seems that there were some issues during the implementation or evaluation.

Response: Thank you very much. As highlighted in Tables 3-8, we improved the results of SVM by conducting more experiments with changing kernel functions of SVM.  However, in some cases performance of SVM is not good enough due to the nature of the cancerous microarray datasets that have few samples with too many features. This is one of SVM disadvantages, which is SVM likely to give poor performance If the number of features is much greater than the number of samples.

(d) Did you use k-fold cross-validation?

Response: Yes, we used 10-fold cross-validation. We write one sentence in Section 3.5. Testing and Evaluation of Machine Learning Techniques line 485-486

(e) Many existing works adopted deep learning algorithms and their results were not perfect; why did NB achieve perfect performance?

Response Although neural networks and deep learning algorithms are common machine learning algorithms, it is time consuming and impractical to use with the cancerous microarray datasets that have too many features. They will need many inputs nodes and more nodes in other layers causing too much computational process. 

- The performance of NB before applying the proposed hybrid filter-DE method is not good in most of cancerous microarray datasets used in this paper since they have numerous redundant and irrelevant features that have a detrimental impact on cancer classification results. After applying the proposed hybrid filter-DE, the performance of NB is outstandingly improved since the proposed method contributed to removing the redundant, irrelevant, and noisy features.

Reviewer 2 Report

Comments and Suggestions for Authors

The article is devoted to the development of a two-stage approach to feature selection that ensures high quality of data classification in the field of cancer diagnostics.

Authors should respond to the following comments.

1.    In line 105 they talk about five feature selection methods, while in line 113 and onwards they talk about six.

2.    In formulas (2)-(8), the authors use the same notations for different concepts. For example, "c", "k" (where "k" plays the role of either a lower index under the sum sign or an upper index). This should not be done. In addition, after each formula, the meaning of the variable used should be explained .

3.    IN lines 309-311 there are the phrase "and 𝑠 and 𝑠𝑖 are the number of samples in 𝑆 and 𝑆𝑖, respectively". However, these designations are not present in formula (8) itself.

4.    Is it necessary to describe the differential evolution algorithm in detail?

5.    For Algorithm 1, there is no need to explicitly represent the table rows.

6.    The authors solve a multiclass classification problem on 2 datasets out of 4. In these cases, it makes sense to use the quality metrics for multiclass classification in Section 3.5. Is there a class imbalance problem in the datasets? Is it solvable?

7.    In tables 3-6 some cells are highlighted. It makes sense to provide an explanation under the tables explaining the highlighting.

8.    What are the time costs of using the author's approach?

9.    The authors consider 5 types of classifiers. It is necessary to conduct statistical tests, for example, the Wilcoxon test and compare the classifiers. It would be important to compare the author's results and the results of other authors based on statistical criteria.

Comments on the Quality of English Language

Minor editing of English language is required.

Author Response

Response to Reviewer 2 Comments

Dear reviewer

We would like to express our gratitude to you for your valuable time and efforts you have spent in the review. Please find the detailed responses below and the corresponding corrections highlighted in the re-submitted files. We have carefully considered your suggestions and made some changes. We have tried our best to improve and make changes in the manuscript.

Sincerely Yours

Corresponding Authors

Dr. Arshad Hashmi

Department of Information Systems

Faculty of Computing and Information Technology-Rabigh

King Abdulaziz University

Kingdom of Saudi Arabia

Assoc. Prof. Dr. Waleed Ali

Department of Information Technology

Faculty of Computing and Information Technology- Rabigh

King Abdulaziz University

Kingdom of Saudi Arabia

  1. In line 105 they talk about five feature selection methods, while in line 113 and onwards they talk about six.

Response 1: We are sorry for this mistake. We used six filter feature selection methods, so we corrected this mistake in line 132

  1. In formulas (2)-(8), the authors use the same notations for different concepts. For example, "c", "k" (where "k" plays the role of either a lower index under the sum sign or an upper index). This should not be done. In addition, after each formula, the meaning of the variable used should be explained .

Response 2: Thank you for valuable comment. We recheck formulas (1)-(8) and rewrite all notations properly.

  1. IN lines 309-311 there are the phrase "and ?and ?? are the number of samples in ? and ??, respectively". However, these designations are not present in formula (8) itself.

Response 3: Thank you for valuable comments. We recheck formulas (1)-(8) and rewrite all notations properly.

  1. 4.    Is it necessary to describe the differential evolution algorithm in detail?

Response 4: As recommended, DE is described in detail in subsection 3.3. Differential Evolution Based Feature Selection

  1. For Algorithm 1, there is no need to explicitly represent the table rows.

Response 5: Done as recommended in Algorithm 1  

  1. The authors solve a multiclass classification problem on 2 datasets out of 4. In these cases, it makes sense to use the quality metrics for multiclass classification in Section 3.5. Is there a class imbalance problem in the datasets? Is it solvable?

Response 6: Thank you for valuable comment. The addressing of data imbalances is out of our paper scope. So, in Section Conclusion and future work, we recommend that using resampling techniques or class-weight learning could enhance the proposed method

  1. In tables 3-6 some cells are highlighted. It makes sense to provide an explanation under the tables explaining the highlighting.

Response 7:

Thank you for valuable comment. The gray cells in Table 3-6   are indicating   the smallest number of selected features .

Already explained in LINE No 571 for Table 3, in LINE No 619 for Table 4, in LINE NO 671 related to Table-5    and in LINE NO 725 related to Table-6 .

  1. What are the time costs of using the author's approach?

Response 8: Thank you very much. As recommended, we discussed the analysis of time complexity of the proposed method in  4.6.  Analysis of Computational Time Complexity line 725

  1. The authors consider 5 types of classifiers. It is necessary to conduct statistical tests, for example, the Wilcoxon test and compare the classifiers. It would be important to compare the author's results and the results of other authors based on statistical criteria.

Response 9: We are sorry about this comment. We have 6 proposed methods and 7 previous works. Due to time constraints, we couldn’t conduct statistical test for comparing all methods since it needs more time and experience. Instead, as recommended by reviewer 1, we calculated the percentage improvements by the proposed work and the existing works in subsection 4.7. Comparison of Proposed Hybrid Filter-DE with Previous Works. Then, we added one sentence in the abstract and introduction section that stated the percentage improvements by the proposed work compared to existing works

Round 2

Reviewer 1 Report

Comments and Suggestions for Authors

The authors enhanced the quality of the paper; however, some comments remain unaddressed. Another round of minor revision is thus recommended.
Follow-up comment 1. There is an incomplete sentence/paragraph in the abstract: Please check “compared to the previous works that are 41.43%, 53.66%, 17.53%, 61.70% on Brain dataset, CNS, LUNGs and BREAST,”
Follow-up comment 2. Refer to the journal’s template; the maximum number of keywords is 10.
Follow-up comment 3. Carefully check the spacing “worldwide[2].It can”, “[19, 20-35] ”, “[18] [24]”, “[23-24] [26-27] [33-35]”, “Storn and Price[47]”, etc.
Follow-up comment 4. Regarding the literature review, only the results of a few works were discussed. Please ensure consistency in summarizing the necessary components, i.e., methodology, results, and limitations of the existing works.
Follow-up comment 5. The resolutions of the figures remain poor. Please zoom in on your paper file to confirm that no content is blurred.
Follow-up comment 6. Table 1: In-text citations are needed for each dataset.
Follow-up comment 7. Carefully format and align the numbers of each equation based on the requirement of the journal’s template.
Follow-up comment 8. Table 2: Only one set of scenarios was tested, which was insufficient to confirm the effectiveness of the proposed algorithm.
Follow-up comment 9. Tables 3-9: In-text citations are missing for the existing algorithms. In addition, how did you design each model optimally?

Author Response

Response to Comments of Reviewer 1

Dear reviewer

We would like to express our gratitude to you for your valuable time and efforts you have made in the review. Please find the detailed responses below and the corresponding corrections highlighted in the re-submitted files. We have carefully considered your suggestions and made some changes. We have tried our best to improve and make changes in the manuscript.

Sincerely Yours

Corresponding Authors

Dr. Arshad Hashmi

Department of Information Systems

Faculty of Computing and Information Technology-Rabigh

King Abdulaziz University

Kingdom of Saudi Arabia

Assoc. Prof. Dr. Waleed Ali

Department of Information Technology

Faculty of Computing and Information Technology- Rabigh

King Abdulaziz University

Kingdom of Saudi Arabia

The authors enhanced the quality of the paper; however, some comments remain unaddressed. Another round of minor revision is thus recommended.
Follow-up comment 1. There is an incomplete sentence/paragraph in the abstract: Please check “compared to the previous works that are 41.43%, 53.66%, 17.53%, 61.70% on Brain dataset, CNS, LUNGs and BREAST,”

Response 1: By mistake we repeated the last sentence in abstract. We removed this sentence as recommended.

Follow-up comment 2. Refer to the journal’s template; the maximum number of keywords is 10.

Response 2: As recommended, we reduced the keywords to 10.

Follow-up comment 3. Carefully check the spacing “worldwide[2].It can”, “[19, 20-35] ”, “[18] [24]”, “[23-24] [26-27] [33-35]”, “Storn and Price[47]”, etc.

Response 3: We corrected these things as recommended.

Follow-up comment 4. Regarding the literature review, only the results of a few works were discussed. Please ensure consistency in summarizing the necessary components, i.e., methodology, results, and limitations of the existing works.

Response 4: As recommended, we enhanced the literature review in Section 2. Related Works by adding methodology, results, and limitations of the existing works

Follow-up comment 5. The resolutions of the figures remain poor. Please zoom in on your paper file to confirm that no content is blurred.

Response 5: We did our best to enhance all the figures

Follow-up comment 6. Table 1: In-text citations are needed for each dataset.

Response 6: As recommended, the datasets are cited in Table 1

Follow-up comment 7. Carefully format and align the numbers of each equation based on the requirement of the journal’s template.

Response 7: As recommended, we aligned the numbers of all equations based on the requirement of the journal’s template

Follow-up comment 8. Table 2: Only one set of scenarios was tested, which was insufficient to confirm the effectiveness of the proposed algorithm.

Response 8: As recommended, we put in Table 2 many scenarios that are tested for the crossover rate and population size

Follow-up comment 9. Tables 3-9: In-text citations are missing for the existing algorithms. In addition, how did you design each model optimally?

Response 9:

-The comparison of the proposed methods against the existing algorithms only in Tables 9 and 10, so we put citations of the existing works. In Table 4-7, we did not compare with existing works. In Table 4-7, we only compared the performance of kNN, NB, DT, RF, and SVM with all features, features selected by filter algorithms only, and features selected by the proposed hybrid filter-DE.

- In lines 558-563, Section 3.4. Training of Machine Learning Techniques, we explained how to design and train models optimally

"To optimally design the kNN, NB, DT, RF, and SVM, the best features selected by the proposed hybrid filter-DE feature selection are used as inputs to train these models. In addition, the best settings and parameters used in all classifiers were selected by a trial-and-error basis in order to produce the best results. Furthermore, we train these models using stratified cross-validation that is especially useful for imbalanced datasets, ensuring equal representation in each fold"

Reviewer 2 Report

Comments and Suggestions for Authors

Despite the corrections made by the authors to the article, the following comments remain.

1.    After modification of annotation in lines 43 and 44 appeared extra fragment text: “compared to the previous works that are 41.43%, 53.66%, 17.53%, 61.70% on Brain dataset, CNS, 43 LUNGs and BREAST,”. It needs to be removed.

2.    Both the abstract and lines 168-711 provide percentages of accuracy improvement obtained by the proposed methods and previously known ones. However, the authors do not indicate here in relation to what these and other improvements were obtained. In addition, they refused to use statistical tests, motivating their answer by the fact that “it needs more time and experience”. Statistical tests must be performed! At least the Wilcoxon signed-rank test.

3.    There is no need to write “where” twice in the explanations for formulas, for example, this is what is written in line 315: “where 𝑆𝑣 represents the subset of data S where feature A has the specific value v”. Check the whole text!

4.    The authors did not respond fully to the comment: “The authors solve a multiclass classification problem on 2 datasets out of 4. In these cases, it makes sense to use the quality metrics for multiclass classification in Section 3.5. Is there a class imbalance problem in the datasets? Is it solvable?

In the case of 4 classes, it is advisable to use quality metrics for multi-class classification. The answer to the imbalance problem has not been received either. Although the imbalance can be easily detected by simply counting class labels. If the methods make significant errors on minority class objects, then such methods are useless and even harmful in medical diagnostics! This problem must be solved here and now, and not in the following articles. What errors do the proposed methods make on minority class objects? This must be shown! If we mistakenly consider sick people to be healthy, then this is very bad! And usually sick people are the minority class!

Comments on the Quality of English Language

Minor editing of English language is required.

Author Response

Please refer to the PDF file ( Response to Comments of Reviewer 2) since there are figures and a table 

Round 3

Reviewer 2 Report

Comments and Suggestions for Authors

Notes.

1.    The sentence This research did not address computational complexity and model validation techniques” (lines 193 and 194) repeats meaning the sentences “In addition, this study did not address computational complexity and model validation techniques” (lines 196 and 197).

2.      It is necessary to detail the errors by classes in the datasets under consideration, especially in the Lung dataset, which is significantly unbalanced. If the classifier makes many errors on objects of the minority class, then this is very bad even if it has high metric values.

3.    In section 3.5, formulas for multiclass classification metrics must be specified. In response to the comment, the authors write about the specificities of calculating various metrics in the case where there are many classes. This must be reflected in the article.

Comments on the Quality of English Language

Minor editing of English language is required.

Author Response

Response to Comments of Reviewer 2

Dear reviewer

We would like to express our gratitude to you for your valuable time and efforts you have made in the review. Please find the detailed responses below and the corresponding corrections highlighted in the re-submitted files. We have carefully considered your suggestions and made some changes. We have tried our best to improve and make changes in the manuscript.

Sincerely Yours

Corresponding Authors

Dr. Arshad Hashmi

Department of Information Systems

Faculty of Computing and Information Technology-Rabigh

King Abdulaziz University

Kingdom of Saudi Arabia

Assoc. Prof. Dr. Waleed Ali

Department of Information Technology

Faculty of Computing and Information Technology- Rabigh

King Abdulaziz University

Comments and Suggestions for Authors

  1. The sentence “This research did not address computational complexity and model validation techniques” (lines 193 and 194) repeats meaning the sentences “In addition, this study did not address computational complexity and model validation techniques” (lines 196 and 197).

Response 1: Thank you very much. We removed the repeated sentence in line 196.

  1. It is necessary to detail the errors by classes in the datasets under consideration, especially in the Lung dataset, which is significantly unbalanced. If the classifier makes many errors on objects of the minority class, then this is very bad even if it has high metric values.

Response 2: Thank you very much for your efforts in improving our paper.

As recommended, we did the following actions to respond to this comment:

- In Section 3.5 lines 590-600, we write the equations that define FPR and FNR for each class.

- In Table 7, for Lung dataset, the classification errors in terms of False Positive Rate (FPR) and False Negative Rate (FNR) were calculated for each class in Table 7.

- In lines 771-802, we discussed The FPR and FNR for each class for SVM, NB, KNN, DT, and RF after applying the proposed hybrid filter-DE feature section methods: IG-DE, IGR-DE, CHSQR-DE, CR-DE, GIND-DE and RE-LIEF-DE.

  1. In section 3.5, formulas for multiclass classification metrics must be specified. In response to the comment, the authors write about the specificities of calculating various metrics in the case where there are many classes. This must be reflected in the article.

Response 3: Thank you very much for your efforts in improving our paper. As recommended, in Section 3.5 lines 578-600, we write the equations and more explanation about multiclass classification metrics.
